# Aversive stimulus-tuned responses in the CA1 of the dorsal hippocampus

Albert M. Barth [1,2,3] ✉, Marta Jelitai[1,3], Maria Flora Vasarhelyi-Nagy[1] & Viktor Varga [1] ✉

Throughout life animals inevitably encounter unforeseen threatening events. Activity of principal cells in the hippocampus is tuned for locations and for salient stimuli in the animals' environment thus forming a map known to be pivotal for guiding behavior. Here, we explored if a code of threatening stimuli exists in the CA1 region of the dorsal hippocampus of mice by recording neuronal response to aversive stimuli delivered at changing locations. We have discovered a rapidly emerging, location independent response to innoxious aversive stimuli composed of the coordinated activation of subgroups of pyramidal cells and connected interneurons. Activated pyramidal cells had higher basal firing rate, more probably participated in ripples, targeted more interneurons than place cells and many of them lacked place fields. We also detected aversive stimulus-coupled assemblies dominated by the activated neurons. Notably, these assemblies could be observed even before the delivery of the first aversive event. Finally, we uncovered the systematic shift of the spatial code from the aversive to, surprisingly, the reward location during the fearful stimulus. Our results uncovered components of the dorsal CA1 circuit possibly key for re-sculpting the spatial map in response to abrupt aversive events.

Survival in changing environments with unforeseen threatening events relies upon the ability to store information about the location and circumstances of aversive situations. Accordingly, fearful stimuli were shown to alter activity in the hippocampus[1], the epicenter of the episodic memory circuit, where spatially tuned activity of principal cells i.e. place cells forms the map that guides behavior during navigation[2]. Hence, hippocampal lesion, especially if targeting the dorsal part, impairs the association of spatial (or contextual) information with aversive or rewarding stimuli and results in failure to mount adaptive behavioral response at salient locations[3,4]. Importantly, a predominant manifestation of many psychopathologies, of which post-traumatic stress disorder is a prime example, is the disruption of fear memory formation and recall that leads to the inability of mounting adaptive responses in fearful situations[5]. Many studies demonstrated that the spatial representation of the environment is distorted in the proximity of relevant locations[6,7]. Remapping, i.e. the redistribution of activity in space could be observed after contextual fear conditioning by various aversive stimuli from air puff through electric shock to predatory odors[8–11]. In a recent study, the presentation of a looming fearful object destabilized spatially tuned firing in its proximity[12]. Threat-triggered hippocampal remapping occurs at various timescales from the immediate alteration of spatial tuning to delayed changes observed following multiple exposures to the aversive stimulus[13]. In the aforementioned examples spatially tuned activity was affected by the threatening event. However, a subset of hippocampal pyramidal cells has recently been shown to respond only to a salient (rewarding) stimulus irrespective of where it was delivered i.e. the activity of these neurons followed the stimulus if it had been

[1]Subcortical Modulation Research Group, Institute of Experimental Medicine, Budapest 1083, Hungary. [2]Cerebral Cortex Research Group, Institute of Experimental Medicine, Budapest 1083, Hungary. [3]These authors contributed equally: Albert M. Barth, Marta Jelitai. ✉e-mail: barth.albert@koki.hun-ren.hu; varga.viktor@koki.hun-ren.hu

translocated[14]. In this work, we asked if the hippocampus carries a specific code for threatening events independent of location and how aversive experience alters the representation of the environment. We have discovered a rapidly emerging, location-independent response to aversive stimuli composed of the sequential, coordinated activation of subgroups of pyramidal cells and interneurons scaled by context and stimulus salience. The threat-selective putative pyramidal cells had higher basal firing rate, more probably participated in ripples and amalgamated into co-active ensembles emerging independent from but dramatically strengthened by aversive stimuli. Surprisingly, the threatening stimulus triggered a shift in the position decoded from coactive ensembles from the current location to the place of reward delivery.

## Results

### Air puff activates a subset of both putative pyramidal cells and interneurons in the dorsal CA1

To uncover the mechanism of aversive stimulus coding in the hippocampus we designed a self-rotating disc with 190 cm running path, decorated with a variety of textures, shapes and colors allowing the mice to distinguish path segments (Fig. 1a). Head-fixed mice were trained to run for water rewards administered at the same position in each lap (except in a few sessions, see later). As an aversive stimulus, air puff was delivered at varying locations on the day of recordings (Fig. 1a, b, Supplementary Fig. 1a). Air puff epochs ($n = 6–10$ trials / location) were separated by inter-air puff laps with no stimulation ($n = 10$). The intensity (200 ms, 30 psi) of the puff was titrated for

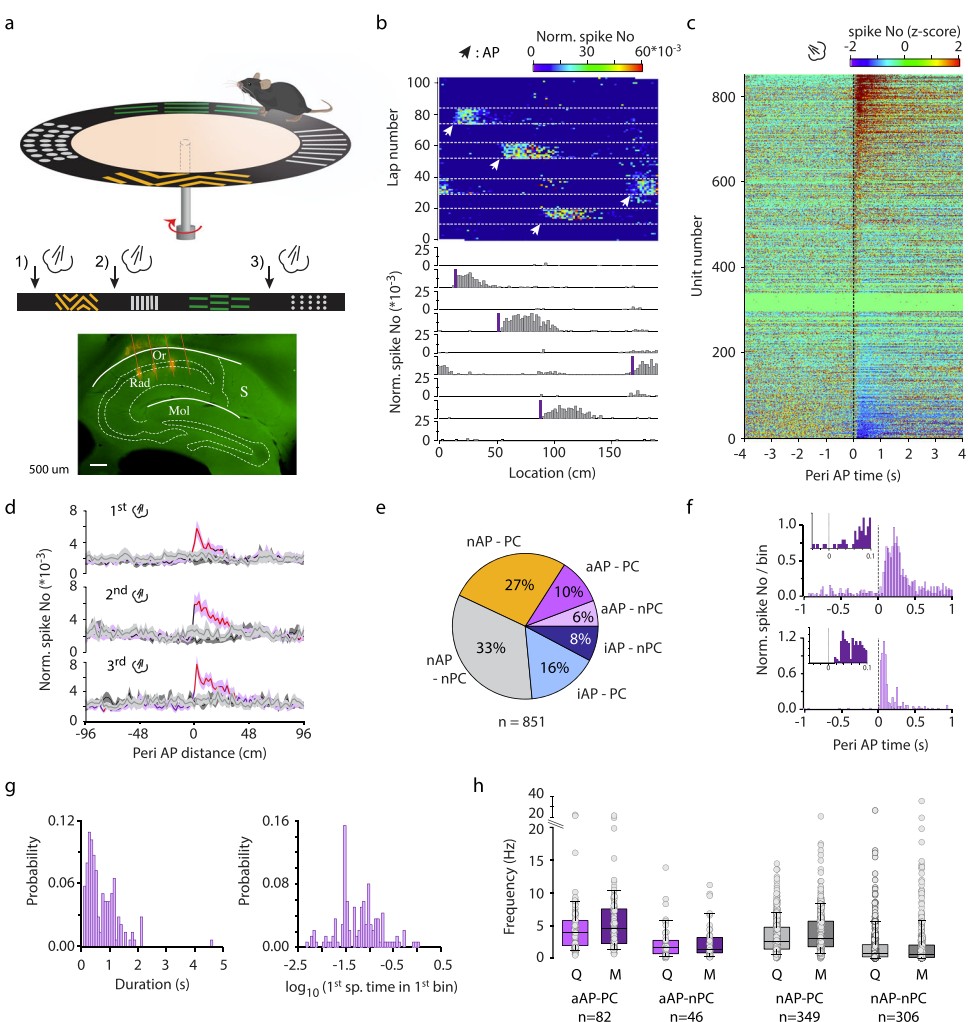

**Fig. 1 | Aversive air puff (AP)-triggered responses of dorsal CA1 pyramidal cells.** **a** Schematics of the recording configuration. Linearized belt indicates the different AP locations during AP epochs. Fluorescent micrograph of a sagittal section showing the tracks of the silicon probe. Mol, molecular layer of the dentate gyrus; Or, stratum oriens; Rad, stratum radiatum; S, subiculum. Scidraw. (2020). mouse running. Zenodo. https://doi.org/10.5281/zenodo.3925913. **b** Representative location-lap number raster (upper plot) and mean tuning curves in no stimulation and AP epochs (lower plots) of an AP-activated pyramidal cell. White arrowheads and violet bars mark AP locations. Light grey dashed lines separate consecutive no stimulation and AP (4) epochs. **c** Peri AP firing histograms of putative pyramidal cells from all session ($n = 851$ units, 18 sessions). **d** Normalized spike number around the AP stimulation calculated in the 1st, 2nd and 3rd AP epochs. Light (pre) and dark (post) grey lines indicate the control epochs, violet line represents the AP epochs, shaded area corresponds to s.e.m. Red line indicates significant differences in spike count between control and AP epochs ($n = 54$, 12 sessions, Wilcoxon signed rank test, p < 0.05). **e** Percentages of AP responsive putative pyramidal cells. aAP, air puff activated; nAP, non-AP responsive; iAP, decreased activity upon air puff stimulation; PC, place cell; nPC, non-place cell. **f** Two representative examples demonstrating the onset of air puff responses (bin size 20 ms). Inset, bin size 5 ms. Note the remarkably rapid response onset of the second aAP-Pyr cell. **g** Distribution of the duration (left panel) and latency (right panel) of AP-evoked activation. Latency was quantified as time from air puff TTL onset to the 1st spike in the 1st significant bin at logarithmic timescale. **h** Firing rate during quiet wakefulness (Q) and movement periods (M). Circles represent individual neurons. 2-way repeated measures ANOVA: Factor A (quiet-mov) F(1779) = 75.94, $p < 10^{-17}$, Factor B (cell groups) F(3779) = 28.65, $p = 1.4*10^{-17}$, interactions F(3779) = 13.69, $p = 1.03*10^{-8}$. Post hoc Tukey test: all cell groups are significantly different from each other except of the aAP-nPC vs nAP-nPC. Box and whiskers correspond to median, quartile and 10-90% range. Note, that we broke the y axis for better visualization. Source data are provided as a Source Data file.

startling the mice without causing a long-lasting freeze of their behavior (Supplementary Fig. 1b).

For sampling the hippocampal circuit, silicone probes (32-256 channel) were inserted into the left ($N = 8$ mice) or both in the left and right ($N = 5$ mice) dorsal hippocampi during isoflurane anesthesia and recording commenced following the return of pre-anesthesia spontaneous behavior. The no stimulation – air puff epochs were repeated one, two or three more times but with different air puff locations. From recordings of 18 sessions 1067 putative single units could be isolated (see Table 1 for session details). Based on well-established classification by firing rate, spike half-width and burstiness[15], 851 units were identified as putative pyramidal cells and 216 units as putative interneurons (Supplementary Fig. 1c). A subgroup of putative pyramidal cells exhibited a sharp increase in activity upon air puff stimulation (16.0 %, $n = 136$ of 851, latency $0.11 \pm 0.165$ s, duration $0.74 \pm 0.59$ s, Fig. 1c). In order to test if the discharge of air puff-activated putative pyramidal cells is locked to the stimulus and not caused by coincidence with spatially tuned activity, location of air puff delivery was shifted multiple times (2 to 4) to non-overlapping segments and their response was recorded to the relocated stimulation. Notably, their firing field followed the air puff location whereas the control level of activity was detected at previous air puff locations thus, we call these neurons Air Puff-activated Pyramidal cells (aAP-Pyr, Fig. 1b, c and Supplementary Fig. 1a; the proportion of cells responding more than one air-puff epochs is shown in Supplementary Fig 1d, e). Importantly, the magnitude and latency of aAP-Pyr response was similar at each air puff location, no rundown or augmentation was observed (Fig. 1d, Supplementary Fig. 1g, h). The first stimulus within an air puff epoch at a given location could not be foreseen as opposed to the subsequent air puffs at the same location. Thus, we tested if there was a systematic change of aAP-Pyr response magnitude and duration within the first and second air puff epochs by comparing the effect of the first, second, third etc. air puffs on aAP-Pyr activity. No significant trend could be detected indicating that expectation based on the association of stimulus and location did not modulate aAP-Pyr response (Supplementary Fig. 2). A further subgroup of pyramidal cells significantly decreased activity (24 %, $n = 200$ of 851, latency $0.63 \pm 0.84$ s, duration $1.58 \pm 0.81$ s, Air Puff-inhibited or iAP-Pyr Fig. 1c,

e). Inhibitory response of spatially modulated iAP-Pyr cells (iAP-PC) usually occurred when the location of air puff coincided with the place field of the neuron (see below). Notably, there was a small set ($n = 8$) of putative pyramidal cells in which long-lasting suppression was preceded by a short latency, rapidly decaying facilitation (latency (facilitation): $0.08 \pm 0.06$ s, duration: $0.22 \pm 0.07$ s, latency (suppr): $0.67 \pm 0.56$ s, duration: $1.28 \pm 0.97$ s). Closer look at the kinetics of the stimulus-triggered activation of aAP-Pyr revealed a wide distribution of response onsets from as short as ~20 ms up to 100 s of milliseconds (Fig. 1f, g $0.11 \pm 0.165$ s). The tactile component of the air puff started about 17 ms after stimulus onset when air reached the animal's mouth detected by the piezo lick sensor (Supplementary Fig. 3a). The duration of the air puff-evoked response ranged from ~100 ms (half of air puff-length) to more than a second ($0.74 \pm 0.59$ s, Fig. 1g). We checked if aAP-Pyr possess air puff-independent but place-bound firing fields (i.e. place fields) and found that a significant number of them ($n = 49$ of 136 or 36% and 5.7% of all 851 Pyr) were active only when an air puff was given (Fig. 1b, e, air-puff activated non-place cells (aAP-nPC)). The rest of the aAP-Pyr ($n = 87$) had a stable place field outside air puff locations (air-puff activated place cells (aAP-PC)) but this place field was unperturbed by the air puff suggesting the simultaneous but independent tuning to the aversive stimulus and location (Fig. 1e, Supplementary Fig. 1a). The almost all or none effect of air puff and the occasional lack of spatial firing fields motivated us to test if aAP-Pyr may be distinguished from air puff nonresponsive cells based on state-dependent activity and position within the pyramidal cell layer. Therefore, we compared their baseline activity to air puff-unaffected pyramidal cells during both quiet and movement periods. In general, spatially modulated pyramidal cells fired faster than non-modulated ones, but within these two groups air puff-activated neurons had significantly higher firing rate than their non-responding peers especially during running (Fig. 1h). Air puff-inhibited pyramidal cells (both place and non-place cells) fired at significantly lower frequency than aAP-PCs whereas air puff-inhibited place cells exhibited higher activity during movement than nAP-PCs (Supplementary Fig. 3b).

The position of pyramidal cells within the pyramidal layer is correlated with state-dependent firing, coupling to oscillations and spatial

**Table. 1 | Summary table presenting details about the experimental conditions**

| Mouse ID (sex, age in days) | Sessions | Genotype | context | No. of AP reloc. | RW. Reloc. | Other stim. | No. of Pyr. Cells | No. of INs | Decoding anal. | Assembly anal. |
|---|---|---|---|---|---|---|---|---|---|---|
| 2293* (m142) | 32_34 | vGat-ires-Cre | Fam. | 4 | yes | | 38 | 1 | | |
| 2304 (m112) | 13_15 | vGlut3-ires-Cre | Fam. | 4 | yes | | 18 | 4 | | |
| 2305 (m122) | 19_22 | vGlut3-ires-Cre | F + N | 4, 3 | yes, no | | 16 | 15 | | |
| (m127) | 28_30 | | Fam. | 4 | yes | | 18 | 24 | | yes |
| 2306 (m114) | 13_15 | vGlut3-ires-Cre | Fam. | 4 | yes | | 12 | 1 | | |
| (m122) | 22_27 | | F + N | 4, 4 | yes,yes | | 43 | 8 | | |
| 2309* (m142) | 34_38 | vGat-ires-Cre | F + N | 4, 4 | yes,yes | | 50 | 14 | yes | |
| 2311* (m141) | 22_25 | vGat-ires-Cre | F + N | 2, 1 | –, – | | 41 | 6 | | |
| 2312* (m138) | 17_19 | vGlut3-ires-Cre | Fam. | 4 | yes | | 53 | 7 | yes | yes |
| (m139) | 31_34 | | F + N | 4, 2 | yes, – | | 27 | 6 | yes | |
| 2313* (m149) | 18_21 | vGlut3-ires-Cre | Fam. | 3 | – | | 50 | 12 | | yes |
| 2339 (m87) | 13_15 | vGlut3-ires-Cre | Fam. | 2 | – | TS | 108 | 20 | yes | yes |
| 2341 (m74) | 3_5 | vGlut3-ires-Cre | Fam. | 2 | – | TS | 16 | 10 | yes | |
| 2356 (m117) | 7_12 | vGlut3-ires-Cre | Fam. | 2 | – | TN | 34 | 14 | | |
| 2365 (m70) | 14_19 | vGlut3-ires-Cre | Fam. | 2 | – | TS | 156 | 20 | yes | yes |
| (m78) | 27_33 | vGlut3-ires-Cre | Fam. | 2 | – | TS, TN | 54 | 16 | yes | |
| 2366 (m62) | 3_8 | vGlut3-ires-Cre | Fam. | 3 | – | TN | 91 | 30 | yes | yes |
| (m72) | 17_21 | vGlut3-ires-Cre | Fam. | 2 | yes | TS, TN | 26 | 8 | yes | |

*The median raphe nucleus of these animals were injected with AAV viruses. 2293, 2309, 2311 mice were injected with AAV2/5_CAG_Flex_ArchT_GFP (VirusVectorCore) 2312 and 2313 were injected with AAV2/5_EF1a_DIO_hChR2_eYFP (VirusVectorCore). Importantly, these mice were not stimulated before the recordings used in this study.

and context-tuning[16], therefore we tested if aAP-Pyr neurons occupy a different niche within the pyramidal layer than other pyramidal cells. Based on the position of the recording channel with the highest amplitude spike waveform relative to the channel registering the highest power ripple, the location of every unit could be estimated and mapped onto the pyramidal layer. We could localize aAP-nPCs to the middle of the pyramidal layer whereas aAP-PCs were the most superficial-biased in our putative pyramidal cell sample although this difference was not significant (Supplementary Fig. 1i).

Next, we analyzed the effect of air puff on interneurons (Fig. 2). Spatially biased firing of interneurons often reflects the place preference of their place cell inputs. Therefore, we asked if air puff-

activated pyramidal cell activity is inherited by interneurons. First, we have revealed robust, short latency, variable length facilitation in a substantial proportion (55/216) of putative interneurons (latency $0.16 \pm 0.33$ s, duration $0.58 \pm 0.56$ s, Fig. 2a-f). Remarkably, the onset of interneurons' responses largely overlaps with that of the rapidly responding aAP pyramidal cells (Fig. 2e). In addition to facilitation, robust suppression of activity could also be observed in a subset of interneurons ($n = 68/216$, latency $1.12 \pm 1.22$ s, duration $0.74 \pm 0.73$ s, Fig. 2a, b, c, f) and we could identify biphasic responses i.e. short latency, rapidly falling facilitation before the long-lasting suppression as well ($n = 15/216$, latency (facilitation) $0.06 \pm 0.09$ s, duration $0.36 \pm 0.47$ s, latency (suppr.) $0.95 \pm 1.18$ s, duration $1.36 \pm 1.82$ s,

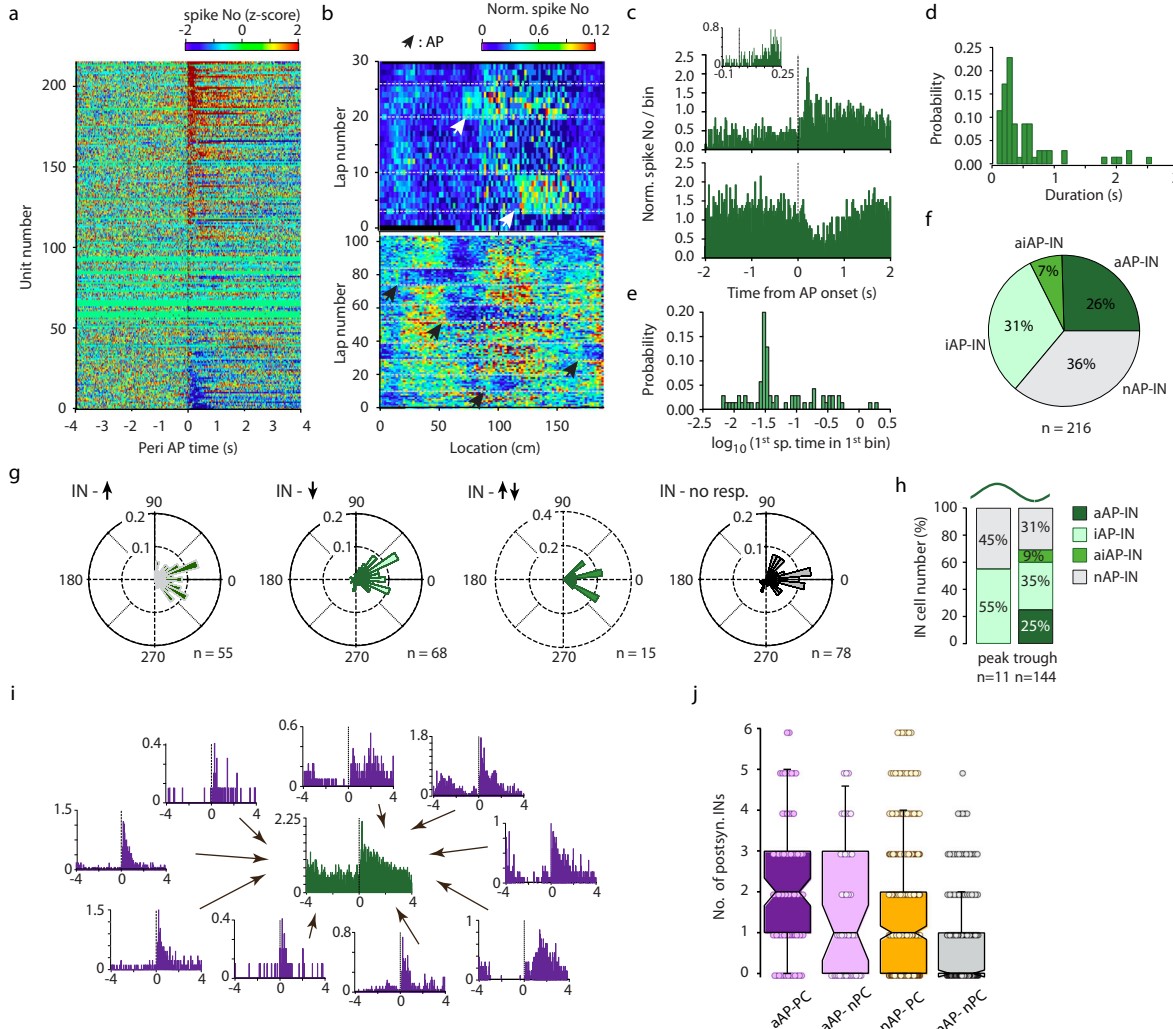

**Fig. 2 | Interneuron response to aversive air puff (AP) stimuli. a** Peri AP firing histograms of putative interneurons from all sessions ($n = 216$, 18 sessions). **b** Representative location-lap number raster of interneurons with increasing (upper panel, 2 AP epoch) and decreasing activity (lower panel, 4 AP epoch), in response to AP. Arrowheads mark AP locations. Dashed lines separate consecutive no-stimulation and air puff epochs. **c** Firing histograms of interneurons, with activation (upper panel) and suppression of activity (lower panel) in response to AP (bin size 20 ms). Inset, bin size 5 ms. **d** Distribution of the duration of AP-evoked activation. **e** Distribution of the latency of air puff-evoked activation quantified as time from air puff TTL onset to the 1st spike in the 1st significant bin at logarithmic timescale. **f** Percentage of putative interneurons with different response types. aAP-IN, increasing activity to air puff ($n = 55$); iAP-IN, decreasing activity to air puff ($n = 68$); aiAP-IN, biphasic response to air puff ($n = 15$); nAP-IN – no response upon air puff stimulation ($n = 78$). **g** Theta phase preference of air puff responsive putative interneurons. aAP-IN, increasing activity to air puff (1.3°; 0.2); iAP-IN, decreasing

activity to air puff (10.4°; 0.3); aiAP-IN, biphasic response to air puff (356.7°; 0.1); nAP-IN – no response upon air puff stimulation (2.4°; 0.4). (mean angle, circular variance). **h** Proportion of air puff responsive and non-responsive interneurons in theta peak and theta trough phase groups of interneurons. **i** Representative example of air puff-activated pyramidal cells converging on an air puff-activated putative interneuron. On each subpanel, peri-air puff firing histogram of the respective neuron is presented. **j** Summary data about the number of postsynaptic interneurons targeted by air puff-activated PC cells (aAP-PC, $n = 87$), air puff-activated non-place cells (aAP-nPC, $n = 48$), non-air puff responsive place cells (nAP-PC, $n = 364$), and nonair puff responsive, non-PC (nAP-nPC, $n = 339$). Kruskal-Wallis test H(3) = 124.53, $p < 0.05$, Dunn-Holland-Wolfe post hoc test: aAP-PC vs aAP-nPC $p < 0.05$, aAP-PC vs nAP-PC $p < 0.05$, aAP-PC vs nAP-nPC $p < 0.05$, aAP-nPC vs nAP-PC $p > 0.05$, aAP-nPC vs nAP-nPC $p < 0.05$, nAP-PC vs nAP-nPC $p < 0.05$. Box and whiskers correspond to median, quartile and 10–90% range. Source data are provided as a Source Data file.

Fig. 2f). We next asked if the air puff-response is correlated with the interneurons' preferred theta phase. First, we determined the phase coupling of each response group. The mean phase of the activated subgroup (aAP-IN) was 1.3°; 0.2 (mean; circular variance), the suppressed subgroup was 10.4°; 0.3, biphasic responders fired at 356.7°; 0.1 and the non-responders preferred 2.4°; 0.4. We also grouped interneurons based on their preferred theta phase into 20° bins and calculated the average z-scored interneuron response for each bin. This analysis revealed that the majority of INs preferred to spike around the trough and both activation and suppression was exhibited by these cells whereas dominantly suppression could be observed in case of the small theta peak-preferring subgroup (Fig. 2g, h and Supplementary Fig. 4a, b[17]). Thus, the aversive stimulus mostly activated trough-preferring interneurons.

Air puff-evoked response of interneurons can be driven by upstream, monosynaptically connected aAP-Pyr cells. Therefore, we attempted to reveal putative monosynaptic connections among aAP-Pyr cells, nAP-Pyr cells and simultaneously recorded air puff-triggered and non-responsive interneurons by analyzing their cross-correlograms (Fig. 2i, Supplementary Fig. 4c, d). A sharp increase of firing probability within a 3-millisecond window following the spikes of a putative presynaptic pyramidal cell indicated a possible monosynaptic connection. We have managed to test connectivity in 12935 pyramidal cell – interneurons pairs and identified 1119 monosynaptic connections. Importantly, aAP-Pyr cells targeted significantly more interneurons than nAP-Pyr neurons (Fig. 2j), whereas aAP-Pyr cells tended to be more prevalent among the presynaptic partners of air puff activated than of nonreacting interneurons (Supplementary Fig. 4e). Thus, aversive stimuli recruited higher level of inhibition via threat-tuned pyramidal cells than a location-locked activity by place cells.

## Amount of spatial information in the environment shapes the hippocampal air puff response

In several situations during navigation spatial cues are not available or can be ignored (navigation in the dark or in cue-poor environments)[18,19]. Accordingly, hippocampal activity is modulated by the amount of spatial information in a context. Thus, we asked if the air puff-triggered responses are altered by exposing the animals to aversive stimulation in a non-spatial versus in a spatial environment. We have constructed a dual context apparatus that allowed us to rapidly switch between a spatial (visuo-tactile cue-decorated disc) and a nonspatial environment (air-supported polystyrene ball) without interrupting the recording (Supplementary Fig. 5). In these experiments air puffs were pseudo-randomly delivered by the experimenter while the animal was sitting or running on the air-supported polystyrene ball or delivered at 2–4 predetermined locations on the cue-rich disc as detailed previously. In the first set of experiments, air puffs were applied first on the polystyrene ball then on the disc and finally on the polystyrene ball again (Fig. 3). To our surprise, we have observed a significantly smaller activation by air puff of pyramidal cells in the nonspatial context: both the magnitude and duration of aAP-Pyr activation were significantly dampened besides the reduction in the proportion of activated pyramidal cells (Fig. 3a-c, z-scored response magnitude: pre-Sph: 5.3 ± 3.9, disc: 10.2 ± 10.5, post-Sph: 4.8 ± 3.2; one-way ANOVA F(2,216) = 13.0, $p = 4.9 \times 10^{-6}$. Tukey's post hoc: pre-Sph vs disc: $p = 1.3 \times 10^{-4}$, disc vs post-Sph: $p = 5.2 \times 10^{-5}$, duration: 0.39 ± 0.42 s, 0.73 ± 0.52 s, 0.37 ± 0.43 s; one-way ANOVA F(2,216) = 15.2, $p = 6.7 \times 10^{-7}$. Tukey's post-hoc: disc vs pre-Sph: $p = 2.3 \times 10^{-5}$, disc vs post-Sph: $p = 1.4 \times 10^{-5}$). In contrast, switching from a cue-rich to a nonspatial context did not lead to significant change in the facilitatory response of interneurons albeit a slight augmentation of activation was detected (Fig. 3d-f, z-scored response magnitude on pre-Sph: 4.7 ± 1.9, disc: 5.0 ± 2.2, post-Sph: 4.1 ± 0.8, one-way ANOVA F(2103) = 2.0, $p = 0.13$, duration: 0.73 ± 0.97 s, 0.51 ± 0.36 s, 0.62 ± 0.61 s, one-way ANOVA

F(2,103) = 0.7, $p = 0.46$). Thus, air puff-triggered interneurons tended to be modified in the opposite direction by the cue-rich to nonspatial context switch. On the other hand, suppression was augmented and characterized a larger proportion of both pyramidal cells and interneurons in the spatial context (Fig. 3b, e). One may raise that exposing the mice first to the spatial cue-deprived environment would interfere with the air puff-triggered response on the cue-decorated context. To exclude this possibility, we reversed the order of contexts: air was puffed first on the disc (spatial) followed by the ball and then by a final disc session but with a changed cue-configuration (novel context). We observed that the air puff-response depended on the type of environment and not on the order of contexts in case of both pyramidal cells and interneurons (Supplementary Fig. 6,7): higher activation still characterized air puff-pyramidal cells in spatial vs. non-spatial context. Augmentation of the response was also observed in case of the air puff-elicited reduction of activity of all suppressed neurons. Notably, we detected a slight but non-significant decrease of air puff effect in the novel versus familiar spatial context (Supplementary Fig. 6,7). Thus, the air puff-elicited response is strongly modulated by spatial information in the environment.

## Effect of stimuli with different salience and valence on AP-cell activity

Spatially tuned activity of hippocampal place cells is thought to be controlled by multimodal sensory input patterns[20]. Tuning to various sensory modalities as well as to the value of stimuli were also reported[14,21–23]. Therefore, we aimed to test if aAP-Pyr cells are sensitive to sensory stimuli irrespective of salience or alternatively, these neurons may be specifically triggered by salient events. To this end, we delivered a highly aversive tail shock (TS) and / or a neutral tone stimulus while the mice were running on the cue-decorated disc (Supplementary Fig. 8). In response to a short tail shock (2*2 ms, 1 mA) a robust but differential effect compared to that of the air puff was observed (Fig. 4a, b). Of putative pyramidal cells 7.7 % were reactive to both air puff and TS, 18.2 % were affected only by the air puff, 23.8 % were TS-only and 50.3 % were non-TS / non-air puff pyramidal cells (Fig. 4c). Next, we delivered a neutral stimulus i.e. pure tone (7.5 kHz, 200 ms) and recorded the response of 273 putative pyramidal cells of which 46 were aAP-Pyr. None of the pyramidal cells, either air puff activated or unaffected, responded to the tone (Fig. 4a). These results suggest that aAP-Pyr cells only partly overlap with TS-units whereas a neutral stimulus not associated with any salient event fails to affect pyramidal cell activity.

Finally, we asked how aAP-Pyr cells are affected by a salient stimulus with positive valence i.e. a reward. In order to detect reward-locked activity we shifted reward location (Fig. 4d). We identified 10 cells (from 157 putative pyramidal cells, 5 sessions) exhibiting significant modulation by reward-increasing firing either right before or after reward delivery ($n = 5$ and 5 neurons, respectively, Fig. 4e). Notably, none of the reward-modulated neurons were significantly activated by air puff thus, no overlap between aAP-Pyr cells and reward-responsive neurons could be observed (Fig. 4e). Additionally, suppression of pre-reward activated cells following both reward and air puff has been detected.

## Coupling of aAP-Pyr cells to the main hippocampal oscillations

Air puff pyramidal cells have stimulus-linked firing fields resembling the spatially tuned neurons' place fields. Place cells fire at progressively earlier phases while the animal is traversing the neuron's place field. This theta phase precession endows place cells with a phase code and enables the formation of coding sequences composed of co-active place cells with shifted theta phase[24]. Additionally, theta timescale compression of information by place cell sequences may be crucial for guiding spatial behaviors[25]. Therefore, we asked if phase precession could be observed in case of aAP-Pyr cells' stimulus-elicited firing? To address this question, circular-linear regression was performed on air

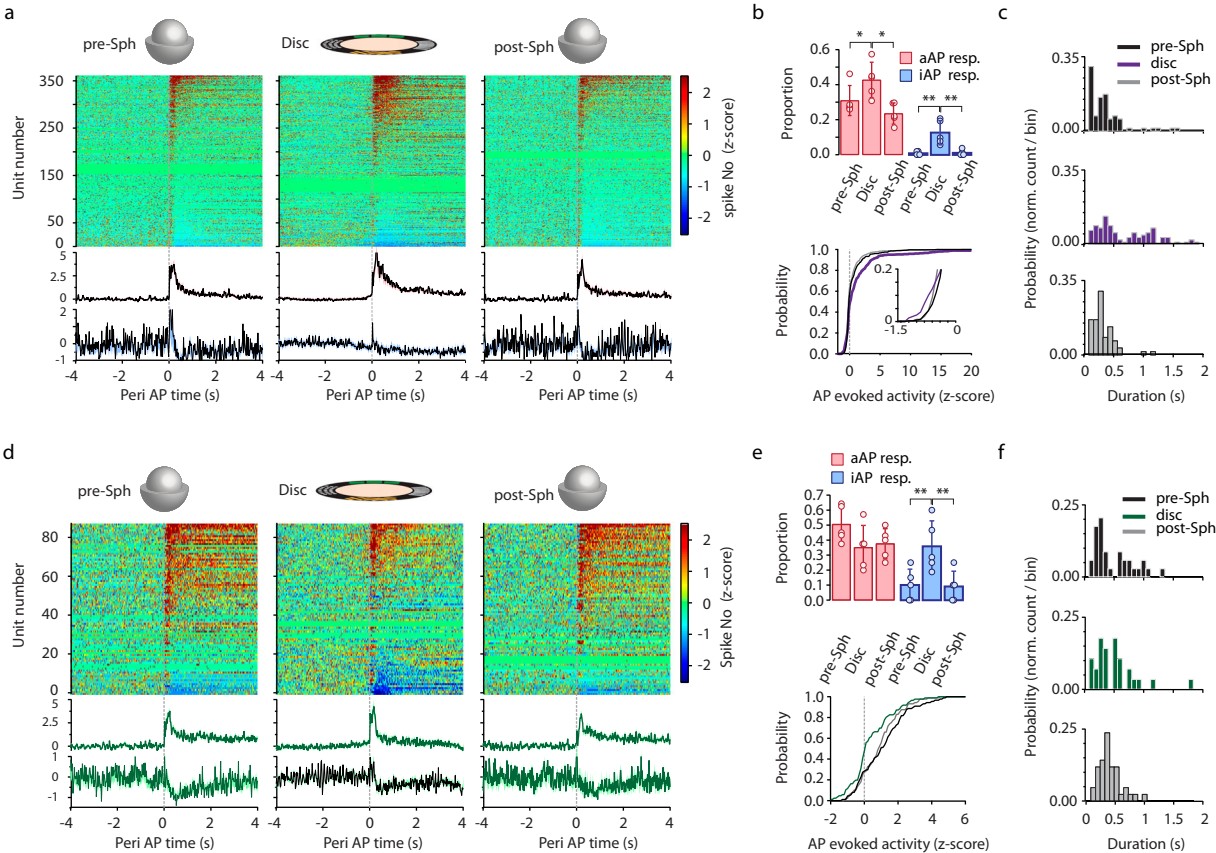

**Fig. 3 | Amount of spatial information differentially modulated the effect of aversive stimulation on pyramidal cells versus interneurons. a** Peri AP firing histograms of putative pyramidal cells (*n* = 361 from 5 sessions) on predisc sphere (pre-Sph), disc and postdisc sphere (post-Sph). Average peri-AP firing histograms of significantly activated and suppressed pyramidal cells are shown below the histograms, shaded areas correspond to s.e.m. **b** Proportion of AP-activated pyramidal cells (upper panel, 5 session, mean ± SD) on pre-sphere, disc and postsphere. Presphere: 0.31 ± 0.086, disc: 0.43 ± 0.10, postsphere: 0.23 ± 0.06; one-way-ANOVA F(2,12) = 6.6, *p = 0.012, Tukey's post hoc test: disc vs. pre-sphere: *p* = 0,11, disc vs. postsphere: *p* = 0.009. Suppressed pyramidal cells on pre-sphere: 0.007 ± 0.01, disc: 0.12 ± 0.07, postsphere: 0.010.016; one-way-ANOVA F(2,12) = 13.3, *p* = 0.0009, Tukey's post hoc: disc vs prephere: **p* = 0.002, disc vs post-sphere: **p* = 0.002. Lower panel: Cumulative distribution of pyramidal cell response magnitudes on pre-sphere (black line), disc (violet) and postsphere (grey). Inset highlights the increased probability of AP-evoked suppression in pyramidal cells' activity on the disc. Kolmogorov-Smirnov test pre-Sph vs disc, **p* = 0.003; disc vs post-Sph **p* = 6*10⁻⁶ **c** Distribution of the duration of AP-evoked activation of pyramidal cells on pre-sphere (black, upper panel), disc (violet, middle panel) and postsphere (grey, lower panel). For visualization purposes time axis was truncated at 2 s also at (**f**) panel. **d** Peri AP firing histograms of putative interneurons (*n* = 88 from 5 sessions) on presphere (left), disc and post-sphere (right). Average peri AP firing histograms of significantly increased and suppressed interneurons are shown below the histograms, shaded areas correspond to s.e.m. **e** Proportion of air puff-activated interneurons (upper panel, 5 session, mean ± SD) on presphere: 0.50 ± 0.12, disc: 0.35 ± 0.15 and postsphere: 0.38 ± 0.10; one-way-ANOVA F(2,12) = 2.2, *p* = 0.15. Suppressed interneurons on presphere: 0.10 ± 0.11, disc: 0.36 ± 0.17, postsphere: 0.09 ± 0.10; one-way-ANOVA F(2,12) = 6.8, *p* = 0.01, Tukey's post hoc: disc vs pre-sphere: **p* = 0.021, disc vs postsphere: *p* = 0.018. Lower panel: Cumulative distribution of air puff-responsive putative interneurons' response magnitude on presphere (black line), disc (green) and postsphere (grey). Kolmogorov-Smirnov test pre-Sph vs disc, **p* = 0.0006; disc vs. post-Sph ** *p* = 0.0007 (**f**) Distribution of air puff-response durations of interneurons on presphere (black, upper panel), disc (green) and post-sphere (grey, lower panel). Source data of panel **b** and **e** are provided as a Source Data file.

puff responses. About one-third (46 out 136 cells) of aAP-Pyr cells exhibited significant theta phase precession (examples on Fig. 5a). Next, we asked to what extent the phase range covered by the air puff-elicited spikes overlaps with that of place field spikes? Answer to this question, would give us a clue if the aversive event occupies a designated temporal position in the hippocampal population code. We have found that aAP-Pyr cells started firing on the early ascending and stopped on the late descending phase close to the trough of the theta cycle thus, air puff spikes covered smaller phase range than place field spikes (Fig. 5a-c). These data prove that the response evoked by the aversive event may be integrated into hippocampal coding sequences and as such may modify the population code (see the last section of the Results).

We next compared the ripple coupling of aAP-Pyr subgroups to non-responding pyramidal cells. We utilized two measures: the proportion of ripples on the disc during which a given neuron (aAP-nPC and aAP-PC versus nAP-PC and nAP-nPC) fired relative to all ripples and

the number of ripple-coincident spikes emitted by the analyzed neuron and [26] normalized to spike number in immobile periods. We found that aAP-PC had the highest probability of ripple participation followed by nAP-PCs whereas aAP-nPC activity co-occurred with ripples with higher probability than firing of nAP-nPCs (Fig. 5d, e, Supplementary Fig. 3c). Moreover, both subgroups of aAP-Pyr cells emitted more spikes during ripples relative to quiet periods than nAP-Pyr (Fig. 5f). Since ripple-coupled reactivation of units representing features of the environment is thought to be a key mechanism of forming long-lasting memory traces, this observation implies higher probability of consolidation of aversive events relative to neutral information[27].

## Modulation of non-air puff pyramidal cell activity by air puff

Next, we turned our attention to exploring the effect of air puff on non-air puff pyramidal cells. The remapping of place cells was reported to have been triggered by aversive stimuli. We asked how a pre-existing

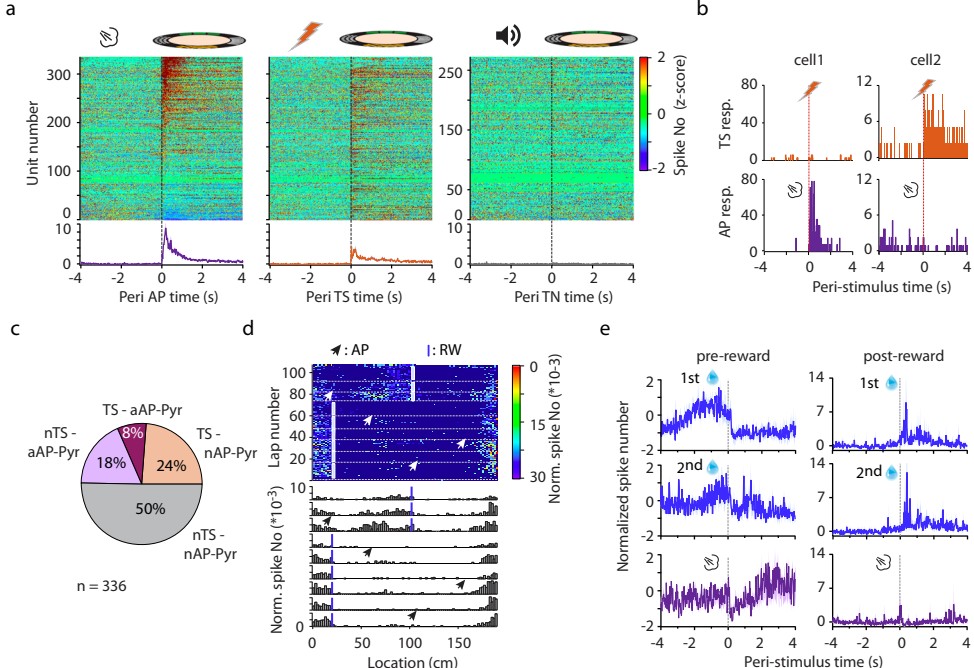

**Fig. 4 | Selectivity of air puff-responsive pyramidal cells to the type and saliency of the stimulus. a** Peri air puff (AP, $n = 336$, 5 session, left panel), peri tail shock (TS, $n = 336$, 5 session, middle panel) and peri tone (TN, $n = 273$, 4 session, right panel) firing histograms of putative pyramidal cells and average histograms of significantly activated cells. Shaded areas correspond to s.e.m. **b** Representative responses of two cells, cell1 responded to air puff and not to tail shock, cell2 responded to tail shock and not to air puff. **c** Percentages of AP and TS responsive putative pyramidal cells ($n = 336$, 5 session). nTS-aAP-Pyr, $n = 61$ air puff-activated but tail shock non-responsive; TS-aAP-Pyr, $n = 26$ both TS and AP activated putative pyramidal cell; TS-nAP-Pyr, $n = 80$ tail shock responsive but air puff nonresponsive putative pyramidal cell; nTS-nAP-Pyr $n = 169$ nonresponsive cell to either tail shock

or air puff. **d** Representative location-lap raster (upper plot) and mean tuning curves during no stimulation and stimulation epochs (lower plots) of a reward (RW) cell. Blue bars mark reward locations, arrowhead and violet bars mark air puff locations. White dashed lines separate consecutive control and air puff (4) epochs. Peri-RW firing histograms in control and air puff epochs are plotted below the location-lap number raster. **e** Average peri-reward (1st reward upper panel, 2nd reward middle panel) and peri-air puff (lower panel) firing histograms of putative pyramidal cells exhibiting significant activation before ($n = 5$ from 10 sessions, left panel) or after reward delivery ($n = 5$ from 10 sessions, right panel). Shaded areas correspond to s.e.m. Markanday, Akshay. (2020). Reward Water Drop. Zenodo. https://doi.org/10.5281/zenodo.3925935.

place field is modified by a co-located air puff and what happens when air puff occurs outside place fields. We selected place cells ($n = 133$) that received aversive stimulus both inside and outside of their place field and calculated their air puff-evoked alteration of activity (Fig. 6). Overall, in-field firing of the majority of place cells was suppressed if the air puff was co-located with their place field whereas their activity, in most of the cases was not significantly (z-score <2) augmented upon out-field delivered aversive stimuli (Fig. 6a-c). Air puff-evoked inhibition can possibly be caused by the coincidence of aversive stimulus onset with decelerating place cell firing in the second half of place fields. To exclude this possibility, we analyzed the correlation of infield relative position of aversive stimulation and evoked inhibition and found no correlation (Supplementary Fig. 9a, b). Thus, suppression of air puff collocated fields was the consequence of inhibition triggered by the stimulus. In a considerable number of place cells global remapping i.e. change of preferred location was registered as reported previously (dots outside the diagonal on Fig. 6d). On the population level, disappearance, emergence and shifting of place fields were observed (Fig. 6d, Supplementary Fig. 9c). In order to quantify the alteration of place cells' spatial tuning, we compared tuning curve correlations within and across conditions: no stimulation vs. no stimulation (pre-pre, post-post), air puff vs. air puff and no stimulation (pre and post) vs. air puff. This analysis uncovered a significantly lower air puff versus no stimulation tuning curve correlation compared to within condition correlations (Fig. 6e). Notably, significant decorrelation of spatially tuned activity from the first to the last no stimulation epoch (pre vs. post) could also be observed indicating a robust reorganization of the map by consecutive aversive stimulation. Since a

substantial number of place cells lost or gained location-coupled activity because of the air puff, we next focused on stable place cells namely, those bearing spatially tuned activity throughout the session and, compared the spatial distribution of their place fields relative to the air puff-location before and after the aversive stimulation. Extensive remapping of stable place cells could be detected without changing place field propensity along the aversive segment of the path relative to non-air puff segments (Fig. 6d). Peri-air puff place field propensity remained unaltered even if not only stable but also vanishing and emerging place fields were included in the comparison of pre- vs. post-stimulation epochs (Supplementary Fig. 9,e). Thus, despite triggering a dramatic rearrangement of place fields along the whole track, the overall probability of place field occurrence was not altered in any segment of the track.

## Emergence of air puff-coupled assemblies

Transient ensembles of co-active neurons coupled to behavioral events and / or brain states are thought to be the basic building blocks of representations as formulated in the assembly hypothesis[28]. Within such neuron coalitions, various types of information defining an episode are amalgamated and later consolidated to memories that guide future behavioral responses[29]. Thus, we asked if aAP-Pyr cells aggregate into air puff-coupled assemblies and whether these appear only during the aversive stimuli or can also be detected independent from the threatening event. We selected 6 sessions in which a sufficiently large number of pyramidal cells were recorded simultaneously (range of pyramidal cell number: 18 – 156) thus, we could expect high enough yield of assembly detection. In

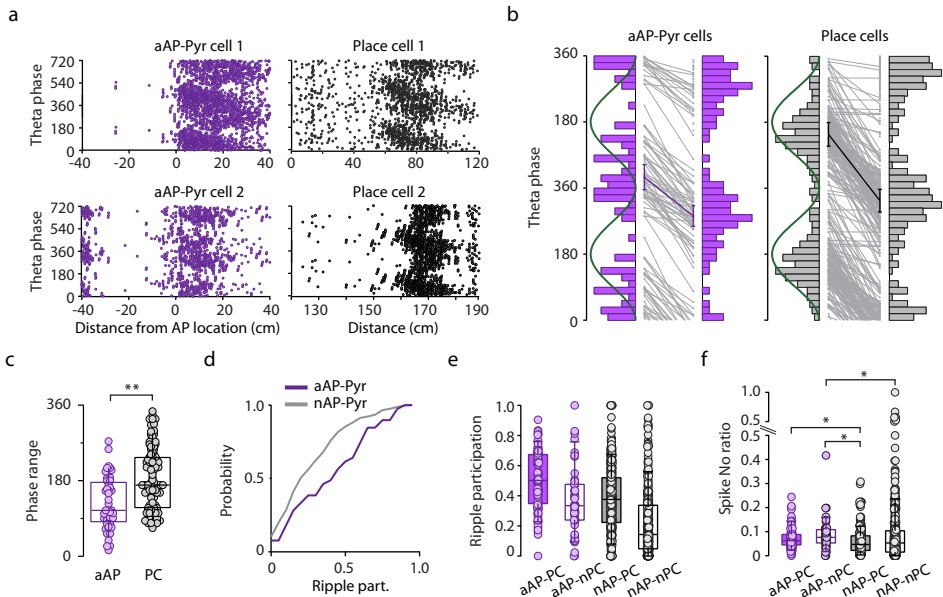

**Fig. 5 | Phase precession and ripple participation of aAP- and nAP-Pyr cells.**
**a** Theta phase precession of two representative aAP-Pyr cells (left panels) and two representative place cells (right panels). Dots mark individual spikes. **b** Paired plots showing individual phase ranges of theta phase precession for aAP-Pyr cells exhibiting phase presession (left panel) and place cells (right panel). Left and right sides of the diagrams are corresponding to start and stop phase distributions, respectively. Two consecutive theta cycles are presented for illustrative purposes. Left side represents start phase and right side represents stop phase. Start phase: aAP-Pyr cells' vs. place cells' 29.6 ± 34.1° vs. 146.9 ± 32.2°, Watson-Williams-test $p < 10^{-3}$. Stop phase: a-AP-Pyr cells' vs. place cells' 283.8 ± 28.3° vs. 326.1 ± 30.9°, Watson-Williams-test $p = 7.8 \times 10^{-4}$. **c** Phase range of aAP-Pyr cells exhibiting phase precession ($n = 46$, aAP) and nAP-PC cells ($n = 130$, PC). Phase range, aAP cells: 126.2 ± 9.0° vs. place cells: 183.4 ± 7.2° Wilcoxon-Mann-Whitney-test **$p = 1.38 \times 10^{-5}$.

**d** Cumulative distribution of ripple participation of aAP-Pyr cells (violet line) and air puff-nonresponsive pyramidal cells (grey line). Kolmogorov-Smirnov test: p = 0.004. **e** Summarized ripple participation of aAP-PC cells ($n = 82$), aAP-nPC cells ($n = 46$), nAP-PC ($n = 349$) and nAP-nPC cells ($n = 306$), Kruskal-Wallis test H(3) = 145.7, $p < 0.005$, There is significant difference between each group except of aAP-nPC vs. nAP-PC with Dunn-Holland-Wolfe post hoc test. **f** Spike number ratio (total number of spikes during ripple / spike numbers during quiet wakefulness) of an aAP-PC cells ($n = 82$), aAP-nPC cells ($n = 46$), nAP-PC ($n = 349$) and nAP-nPC cells ($n = 300$), Kruskal-Wallis test H(3) = 17.6, $p < 0.005$. Post hoc Dunn-Holland-Wolfe test, significant differences are labeled by *, note that for better visualization y axis was broken. On panels **c,e,f** box and whiskers correspond to median, quartile (25−75%) and 10−90% range. Source data of panel **c,e** and **f** are provided as a Source Data file.

these sessions, we could identify 5 to 26 putative pyramidal cell assemblies by the algorithm described in Lopes-dos Santos et al[30]. (Supplementary Fig. 10a, b). Some of these had a clear place preference. Importantly, we could find assemblies linked specifically to the aversive stimuli (7 out of 91, Fig. 7a). To unequivocally identify air puff-coupled ensembles, we also calculated similarity of all detected assemblies to a synthetic air puff-pattern with aAP-Pyr cell weights set so as the sum of squares of weights equaled to 1 (see Methods and Supplementary Fig. 10c, d). A similarity threshold was defined by permuting cell identifiers and repeating the calculation 1000 times. Altogether, 7 significant air puff-linked assemblies could be identified in 6 sessions (5 sessions with 1 and 1 session with 2). By examining the neurons forming the identified assemblies, the prevalence of aAP-Pyr cells within these aversive coactivity patterns was found to be significantly higher than in other, non-air puff-locked assemblies (Fig. 7b). In turn, the weight of aAP-Pyr cells reflecting their contribution to an air puff-assembly was significantly higher than the weight associated with other assemblies (Fig. 7b). Consequently, air puff-assemblies not only collected aAP-Pyr cells, but these units were dominant members (Fig. 7c). However, emergence of co-active ensembles of neurons can simply be the consequence of co-triggering and thus "artificially" synchronizing the stimulus-sensitive units. Therefore, we asked if similar assemblies collecting aAP-Pyr cells could be detected independent from the aversive stimulus. To address this question, we identified assemblies in non-air puff epochs of sessions and then calculated a similarity index between these and the above-mentioned synthetic air puff pattern (Fig. 7c, d, Supplementary Fig. 10b, c). Strikingly, even before the delivery of the first air puff, assemblies highly similar to the synthetic

air puff pattern could be identified on the running disc. These assemblies were spatially scattered and not locked to any location or behavioral event in no stimulation or post-air puff epochs (Fig. 8a). The air puff-independent emergence of air puff assemblies raised the intriguing possibility that these ensembles may be pre-formed without any external influence even before the start of the session and then reactivated by the salient event. In order to test this possibility, we repeated the assembly detection in both the pre-session and post-session non-spatial epochs (mice on the polystyrene ball) and then calculated the similarity between the detected assemblies and the synthetic air puff pattern. We were able to identify pre- and post-session ensembles resembling the air puff assemblies but the weight of aAP-Pyrs in these assemblies was clearly lower in most of the cases than in similar air puff assemblies isolated during the air puff-session (Fig. 7e). Although conditions i.e. pre vs. disc and disc vs. post had a significant effect on weight of participating aAP-Pyr cells, pairwise comparisons (pre vs. disc or disc vs. post) failed to reveal significant difference probably because of large variance and low number of assemblies (n = 4 for this analysis). Further analysis of air puff-independent emergence of air puff assemblies uncovered significantly higher activation strength while the mice were immobile compared to running (Fig. 8c-e). During immobile states sharp wave ripples dominate hippocampal activity. Thus, we analyzed if air puff assemblies preferentially appear during ripples outside air puff-epochs and observed strong coupling between ripples and these assemblies i.e. a sharp increase in assembly reactivation strength around ripple peaks. However, coupling strength of air puff- versus non-air puff ensembles fell into the same range (Fig. 8f). Notably, the activation strength of air puff assemblies elevated gradually

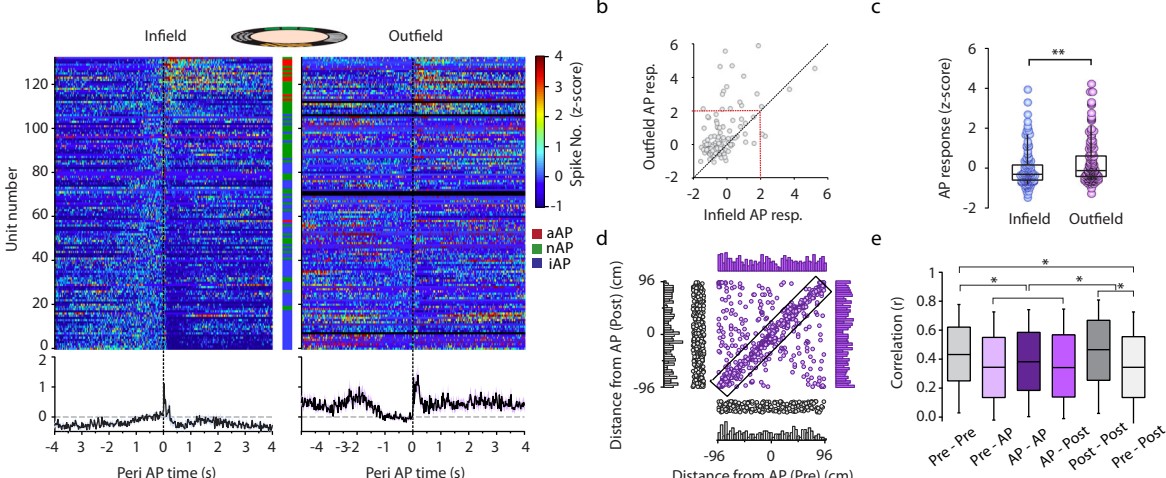

**Fig. 6 | Aversive air puffs suppress place cell spiking within collocating place fields and augments out of field activity while triggers remapping. a** Peri air puff z-scored firing histograms of putative pyramidal cells if air puff was given inside place field (Infield, left panel, *n* = 133) or outside place field (Outfield, right panel, *n* = 133). The color bar between the firing histograms indicates the identity of the corresponding cell: red, aAP, air puff-activated place cell; green, nAP-PC, air puff-non-responsive place cell; blue, iAP, air puff-inhibited place cell. Below the PSTH plots are the averages of the corresponding matrices, shaded error bars correspond to s.e.m. **b** Z-scored outfield air puff responses as a function of infield air puff responses of putative pyramidal cells (upper panel). Circles represent putative pyramidal cells. Red line marks z-score value of 2 (2 SD). **c** Summarized data showing z-scored infield and outfield air puff responses (infield: 0.11 ± 1.63, outfield: 0.28 ± 1.1 Wilcoxon signed rank test **p = 0.0005, *n* = 133 units; middle, box and whiskers correspond to median, quartile and 10-90% range). **d** Place field distances

from the location of air puff stimulus before the stimulus (Pre) versus after the stimulus (Post). Place fields along the diagonal correspond to stable place fields, off diagonal circles are shifted place fields (remapped). Circles along the x axis are the vanishing and along the y axis are the emerging place fields. Air puff responses at different locations are pooled together. Grey histograms show the distribution of emerging, vanishing fields, violet histograms correspond to stabile and remapping place fields. **e** Correlations of place maps (*n* = 364 nAP-PC, 3 AP locations) calculated between first and second halves of pre stimulation (Pre), air puff stimulation (AP) and post stimulation epochs (Post). Kruskal-Wallis test H(5) = 126.6, *p* < 0.005. Dunn-Holland-Wolfe: there are significant differences: * AP-AP vs Pre-Pre; Post-Post and Pre-Post vs Pre-Pre or Post-Post. Box and whiskers correspond to median, quartile and 10-90% range. Source data of panel b-e are provided as a Source Data file.

throughout the recording session as opposed to the lack of strengthening of non-air puff assemblies (Fig. 8a, b). Based on these findings we hypothesize that aversive events reactivate pre-formed assemblies composed of both future air puff-triggered and unresponsive or suppressed units and the threatening event then robustly increase the weight of threat-coding cells in these assemblies.

## Aversive stimulus disrupts the population code and shifts the representation to the reward zone

The collective activity of hippocampal neurons tuned for various features of the environment forms a map (O'Keefe and Nadel, Hippocampus as cognitive map, 1978). Accordingly, the animal's position can be reconstructed from the activity of simultaneously recorded place cells. Hence, we asked how an aversive event affects the hippocampal map? We have estimated the position of the animal by implementing a Bayesian decoding algorithm[31] and calculated the decoding error i.e. the difference between the estimated and real position (Fig. 9a, b). Additionally, we only included periods when the animals' running speed exceeded > 2 cm/s in order to exclude confounding patterns coupled to immobility. First, animal's location during no-stimulation laps was reconstructed by decoding the second half of the first no-stimulation epoch (before delivery of the first air puff) based on the first half. Error of this reconstruction was compared to that obtained by decoding the air puff epoch based on the second half of the preceding no-stimulation epoch. This analysis revealed a significant increase of decoding error in a 35 cm segment after the air puff location (decoding error compared to no stimulation: *n* = 9 sessions, Fig. 9b). Next, we aimed to uncover if the disruption of the spatial code is caused by a random deviation of estimated from current location or alternatively, by systematic shift of the representation to a well-defined location. To our surprise, the latter alternative was observed, namely following the air puff, the estimator located the animal at the reward

area (Fig. 9c). We tested this finding against the null hypothesis that the probability of any other location's overrepresentation is similar to what we observed for the reward zone. We performed a Monte Carlo simulation in which the representation of randomly selected locations was compared to the experimental findings. The analysis showed that only the representation of reward location exceeded the level of significance (Fig. 9c, see insets). We intended to determine the contribution of aAP-Pyr cells to this representational shift by removing these neurons from the dataset used for location decoding. Repeating the position estimation without aAP-Pyr cells led to the same outcome i.e. the aversive stimulus shifted the representation from the air puff to the reward location (Fig. 9c). We went further to reveal the major contributors to the systematic representational drift. First, we selected air puff-coincident, reward-coding theta cycles from the first air puff epoch when the aversive stimulus was delivered at the farthest location from the reward. Then, we ranked pyramidal cells based on their spike number during these theta cycles, separated them into quintiles and averaged the tuning curves of pyramidal cells in each quintile (Fig. 9d, Supplementary Fig. 11). We assumed that higher firing rate cells influence more the outcome of position estimation than low activity neurons. Surprisingly, pyramidal cells in the highest activity quintile possessed place fields in the proximity of the reward zone whereas lack of reward-bound enrichment of place fields or even a decreased peri-reward firing probability characterized place cells in lower activity quintiles. Thus, an aversive event distorts the hippocampal map by emphasizing locations with opposite valence to the triggering experience by recruiting place cells with reward-proximal place fields.

## Discussion

Threatening events modify the map of the environment in the hippocampus and connected regions whereby enabling the avoidance of

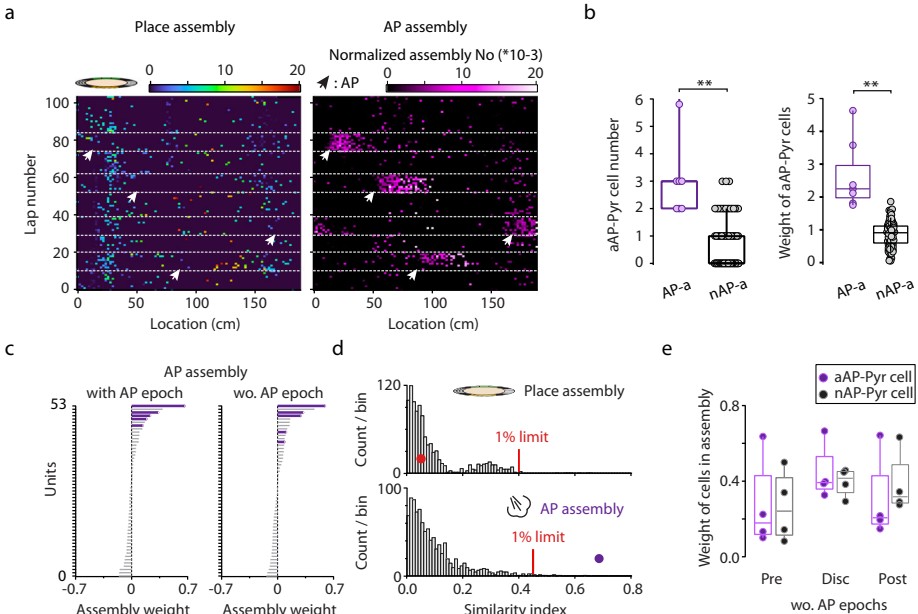

**Fig. 7 | Air puff selective assembly patterns. a** Representative assembly reactivation location – lap number raster with place (left panel) and air puff preference (right panel, from same session as example neuron on Fig. 1b). Arrowhead mark air puff locations. White dashed lines separate consecutive control and air puff (4) epochs. Assembly activation strength are normalized by the total bin count. **b** Summary data of the number of aAP-Pyr cells (left panel, $3 \pm 1.4$ vs $0.7 \pm 0.9$, $p^{**}=1.7 \times 10^{-6}$) and weight of aAP-Pyr cells (right panel, $2.6 \pm 1.1$ vs $0.9 \pm 0.4$, $p^{**}=3.4 \times 10^{-7}$) in air puff assemblies (AP-a, $n = 7$) and in non-air puff assemblies (nAP-a, $n = 84$), for testing significance Wilcoxon-Mann-Whitney test was used. **c** Representative air puff-assembly identified by using the whole session (left panel) or only the non-air puff epochs (right panel). Weights of simultaneously recorded units in the two assemblies arranged in descending order. aAP-Pyr cells are color coded in violet. **d** Similarity of a representative place (upper panel) and air puff

assembly (lower panel) to a synthetic air puff assembly. Histograms correspond to similarity index distributions generated by shuffling unit IDs. Red line indicates similarity threshold (1%) and filled circles mark similarity index of the example place (red dot) and air puff assembly (violet dot). **e** Weight of AP-Pyr cells and non-AP-Pyr cells in air puff assemblies ($n = 4$ from 4 sessions) detected in pre- and post-air puff sessions on the polystyrene ball and in the non-air puff epochs of the air puff session on the disc. For testing significance 2-way repeated measures ANOVA was used. Factor A (aAP – nAP Pyrs), $F(1,6) = 0.006$, $p = 0.94$; for Factor B (Pre, Disc and Post) $F(2,12) = 5.778$, $^*p = 0.017$; Interaction $F(2,12) = 1.21$, $p = 0.33$. There was no significant difference with Tukey post hoc test between Pre, Disc and Post conditions. On panels b,e box and whiskers correspond to median, quartile (25%–75%) and 10–90% range. Source data of panel b,e are provided as a Source Data file.

potentially harmful situations in the future. In this study we uncover the rapidly emerging pattern of neural responses evoked by aversive stimuli in the dorsal hippocampus that ultimately leads to the distortion of the spatial map at fearful places.

The first component of this pattern was the aversive stimulus (air puff)-specific activation of a subset of pyramidal cells with surprisingly short, in most cases < 50 ms latency paralleled by an activity surge of some putative interneurons. Activation was locked to the aversive stimulus, thus when the air puff was relocated, activated cells followed the stimulus without exhibiting lasting change at the preceding stimulus location. Notably, a subgroup of triggered pyramidal cells lacked place fields (aAP-nPC) and even in case of spatially tuned air puff-triggered cells (aAP-PC), the location-locked activity was independent from and unperturbed by the aversive event. An air puff is composed of a high-intensity multispectral sound generated by the solenoid valve immediately followed by the loud outburst of air. Both auditory and tactile information evoke an excitatory response in a substantial proportion of hippocampal neurons with a slightly longer but overlapping latency relative to that reported by us[23,32]. However, rapid, robust activation of both pyramidal cells and interneurons differentiated the air puff response from the effect of neutral stimulation and suggests that strong co-activation of many types of sensors renders a stimulus highly salient. Convergent multimodal sensory input conveying threat-related information would reach the hippocampus possibly via the lateral entorhinal cortex[23] and activate aAP pyramidal cells. These neurons then promptly trigger synaptically connected inhibitory neurons. Notably, the aAP subpopulation targeted more interneurons than non-responsive pyramidal cells. Further excitation of air puff-interneurons may come from subcortical regions such as the

median raphe broadcasting saliency signals to the hippocampal circuitry[33]. Converging local and subcortical excitation may thus efficiently recruit inhibition in response to an aversive stimulus. All activated and majority of suppressed interneurons fired at or around the trough of pyramidal layer theta overlapping with the phase range of dendrite-targeting interneuron subtypes (Oriens-Lacunosum Moleculare or, O-LM and bistratified cells[17]). Notably, O-LM cells are the dominant source of feedback inhibition in the CA1[34] explaining the suppression of co-active place cells around the aversive location by lateral inhibition brought about by threat-triggered pyramidal cells. This transient inhibitory spike would also suppress the formation of new place fields[35] which explains the lack of the overrepresentation of the aversive location. Besides activation, simultaneous suppression of interneuronal activity was also observed raising the possibility that long latency pyramidal cell response was partly caused by disinhibition.

Key questions concern if aversive stimulus-triggered pyramidal neurons are distinct from "ordinary place cells" or other, non-spatially tuned pyramidal cells e.g. reward cells and to what extent these neurons are selective for the type of salient stimulus? Large pool of data collected in recent years demonstrate the diversity of pyramidal cells in hippocampal subregions separable along molecular, anatomical, electrophysiological and functional axes. Firing rate and correlated context-dependent variability subdivides CA1 pyramidal cells into a high-activity group with rigid and less specific spatial tuning and a low activity, plastic, specifically tuned subset[36]. Later neurons were more likely to be coupled to ripples. A correlated classifying dimension corresponds to a deep-superficial position within the pyramidal layer. Superficial pyramidal neurons have lower overall firing rate, higher

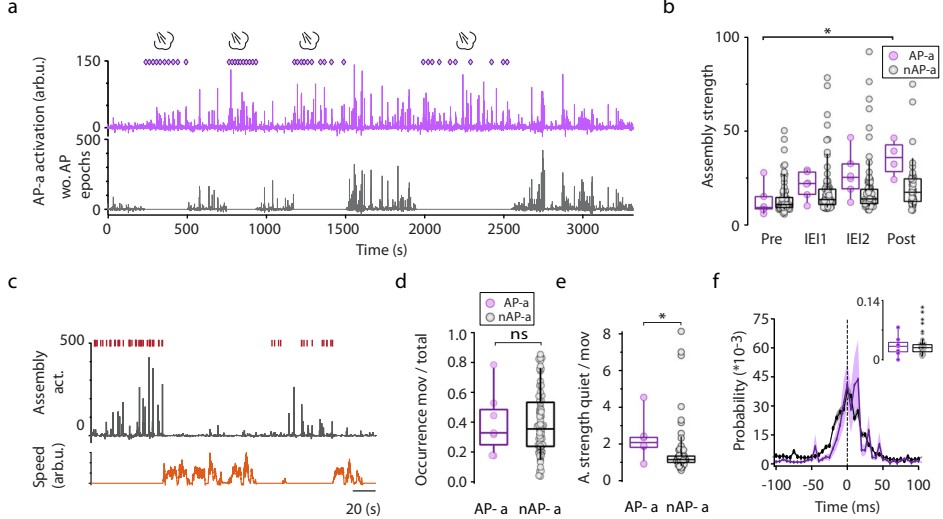

**Fig. 8 | Reactivation of air puff selective assemblies. a** Occurrence of a representative air puff assembly identified by using both AP and non-AP epochs (violet, upper panel) during the session and similar air puff assembly identified by opting out air puff epochs (grey, lower panel) from the same session. Violet rhombi indicate air puff stimulation. For visualization purposes, assembly activation during air puff epochs was removed from the lower panel. **b** Assembly activation strength before air puff stimulus (Pre) between air puff stimuli (3 air puff stimuli, Inter Event Interval 1 and 2 (IEI)) and after air puff stimuli (Post). Violet circles mark air puff assemblies ($n = 6$ from 6 sessions) and grey circles correspond to non-air puff assemblies ($n = 84$ from 6 sessions). (2-way repeated measures ANOVA: Factor A (AP-a – nAP-a), $F_{(1,43)} = 0,627$, $p = 0.433$; for Factor B (Pre, IEIs and Post) $F_{(3,129)} = 8.679$, $p = 2.74*10^{-5}$; Interaction $F_{(3,129)} = 4,784$, $p = 0.003$). There are significant differences between Pre AP-a strength and Post AP-a strength with Tukey post hoc test, $*p = 0.021$. **c** Representative air puff assembly reactivation (grey, middle panel) in comparison to the occurrence of ripples (red, upper panel) and speed (orange, lower panel). **d** Reactivation occurrence of air puff-assemblies

(AP-a, $n = 7$ from 6 sessions) and non-air puff assemblies ($n = 84$ from 6 sessions) during movement. For testing significance Wilcoxon-Mann-Whitney two-sample rank-sum test was used, $p > 0.05$. **e** Ratio of assembly reactivation strength of air puff and non-air puff assemblies during immobility versus movement of air puff and non-air puff assemblies (AP-a, $2.36 \pm 1.2$, $n = 6$ from 6 sessions; nAP-a, $1.49 \pm 1.26$, $n = 84$ from 6 sessions). For testing significance Wilcoxon-Mann-Whitney two-sample rank-sum test was used, $*p = 0.012$. **f** Ripple-triggered average of normalized assembly reactivation strength for both air puff- and non-air puff assemblies, shaded area corresponds to s.e.m. Inset shows the statistically similar normalized reactivation strength of air puff- and non-air puff assemblies in the $\pm10$ ms peri-ripple peak window (AP-a, $0.032 \pm 0.025$, $n = 7$ from 6 sessions; nAP-a, $0.030 \pm 0.02$, $n = 84$ from 6 sessions Wilcoxon-Mann-Whitney two-sample rank-sum test, $p = 0.73$). On panels b,d,e and inset of f, box and whiskers correspond to median, quartile and 10-90% range. Source data of panel b,d,e are provided as a Source Data file.

ripple participation and exhibit better context discrimination as opposed to higher firing rate, lower ripple-coupling and more non-spatial (e.g. reward) and local cue coding of deep neurons[37–39]. Elevated firing rate during movement relative to non-responders, diverging theta phase preference and higher probability participation in ripples imply that aAP-Pyr cells form a separate subpopulation possibly corresponding to the rigid, high firing rate subgroup described previously. Intra-pyramidal layer position corroborated this classification, namely pure aAP-Pyr could be localized to the mid portion of the pyramidal layer whereas mixed aAP-PC were the most superficial subgroup. Thus, the latter cells may combine a rigid place code with an aversive stimulus-coupled variable tuning. However, further studies would be necessary to decide if either group of aAP-Pyr possess distinguishing cell-autonomous attributes or only a unique set of inputs distinguishes them from ordinary place cells. The abovementioned distinguishing characteristics of air puff- and especially mixed selectivity air puff place cells resemble to those of engram place cells[40] which raises the intriguing possibility that these neurons may form the core of reactivable fear memory engrams. Difference in the phase range within theta cycles aAP-Pyr encompass imply that the aversive information these neurons are tuned to have a well-defined but distinct position within the theta-organized coding sequences compared to non-affected place cells. This finding agrees with the assumption that various types of information defining an episode are coupled to different phase ranges of a theta cycle[29]. Disruption of this temporal order may interfere with integration of aversive and spatial information when forming the representation of the fearful event.

A key difference between reward and air puff-triggered pyramidal cells is that activation of the former precedes, consequently predicts the

occurrence of rewards[14]. In contrast, aAP-Pyr cells always followed stimulus onset even though by a very short latency that makes them ideal for triggering, via local inhibition, the reorganization of representation prompted by highly relevant but unpredictably occurring events. Lack of modulation by reward, no response to neutral auditory stimulation and divergent response to a more salient, mildly painful stimulus with the same valence (Tail Shock) support the selectivity of threat-triggered neurons to nonpainful aversive events possibly explained by distinct constellations of inputs activated by the various types of stimuli. Hence, air puff-activated pyramidal cells may be selective not for a given type of event, but for certain input patterns engaged, among others, by air puffs. Differential response to the tail shock can also be because of the divergent subcortical modulation triggered by the tactile versus noxious stimulus. A striking observation was that in the cue-rich configuration, threat-triggered principal cell but not interneuron response had much higher magnitude than in the absence of visuo-tactile information. As opposed to activation, suppression of both pyramidal cells and interneurons was augmented in the cue-rich environment. We could exclude the possibility that habituation or desensitization to the stimulus caused response modulation by testing the mice in both sequences of contexts (nonspatial – spatial – nonspatial and vice versa). A possible explanation can be the higher excitability of principal cells in a spatial context by the converging effect of higher impulse flow through pathways carrying visuospatial information.

Many studies investigated the modulation of place cells' location-locked firing by fearful events and reported a wide range of responses (see the Introduction). Generally, in the proximity of threatening sites, the spatially tuned activity becomes unstable[41]. Here, we observed a clear inhibition of within-place field spiking if the aversive event

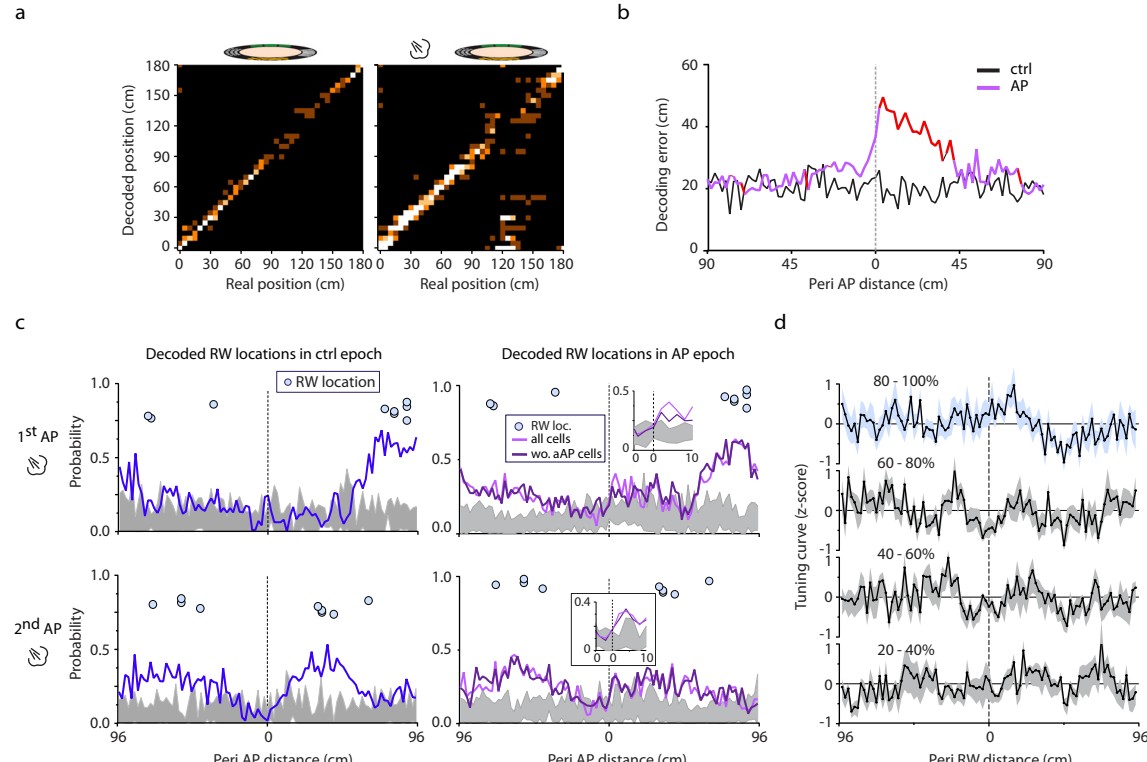

**Fig. 9 | Air puff disrupts the population place code and causes its systematic shift. a** Real versus decoded positions in control (left panel) and air puff (right panel) epochs of a representative session. **b** Decoding error in control (black) and air puff (violet) epochs of all sessions used for position estimation ($n = 9$ sessions). Red indicates significant difference (paired t-test, $p < 0.05$). **c** Probability of estimating the position of mice to be at the reward location in pre-air puff control and first inter-air puff (blue traces upper and lower left plot, respectively) and first and second air puff epochs (violet traces, upper and lower right plot, respectively). Grey area corresponds to repeating 1000 times the estimation of random locations outside the reward zone. Light blue circles mark reward locations. Light violet line corresponds to probability without aAP-Pyr cells. Insets right to the air puff-epoch estimations zoom in on the −5 cm to +10 cm

segment around the start location of air puff-delivery to better show the significantly higher probability of locating the animal to the reward zone following the air puff. Mean probability of decoding to the reward location in the 0–35 cm from air puff delivery: $0.09 \pm 0.14$ in $1^{st}$ no stimulation epoch vs. $0.28 \pm 0.13$ in $1^{st}$ air puff epoch, Wilcoxon signed rank test: *$p = 0.01$; $0.11 \pm 0.09$ in $2^{nd}$ no stimulation epoch vs. $0.25 \pm 0.08$ in $2^{nd}$ air puff epoch, Wilcoxon signed rank test **$p = 0.008$. **d** Z-scored averaged tuning curve of putative pyramidal cells firing during reward coding theta cycles from a 3 sec segment after air puff stimulation. Pyramidal cells were sorted into quintiles based on within cycle firing rate and their tuning curves were averaged in each quintile (shaded areas correspond to s.e.m.). Peri-reward elevation of place field tuning curve is detectable only in the uppermost quintile collecting the most active place cells.

overlapped with a place field. The opposite, i.e. a non-significant elevation of activity could be registered when the threat stimulus arrived at an out-of-field position. New, persisting place fields also appeared albeit surprisingly, only rarely locked to threat locations. These alterations resulted in unchanged place field propensity around the aversive event relative to pre-air puff unstimulated laps. This is contrary to the over-representation of locations associated with rewards in goal-oriented learning tasks[42]. The perturbation caused by the fearful stimulus is also reflected in the temporary disruption of the population place code. Hence, the estimation of the animal's position in the air puff epochs from the combined activity of co-registered place cells during the no-stimulation period led to a sharp elevation of error in position estimation. Surprisingly, this error was not random but systematic: position decoding placed the animal in the rewarded segment while having been exposed to an aversive stimulus. We could exclude the possibility that intermittent immobility triggered by both rewarding and threatening events instantiated correlated patterns that confused the decoder by leaving out immobile and low-speed periods. Additionally, the animals' location was correctly determined to have been at the reward position and never at the air puff-associated places during reward delivery. However, we found that during air puff delivery the place code was dominated by place cells possessing reward-proximal place fields.

The hippocampal population code is composed of transiently emerging co-active ensembles of neurons[28]. These ephemeral neuron

coalitions integrate multiple types of information, and their formation is coupled to locations, events or can be observed during states without external perturbations (e.g. sleep[43],). In our experiments we could observe assemblies linked to locations, to reward delivery and importantly, some of them were tightly coupled to the fearful stimulus. Threat-triggered neurons significantly contributed to the aversive stimulus-locked assemblies. Moreover, ensembles resembling the threat-related assemblies could be detected even before the delivery of the first aversive stimulus as well as in nonspatial contexts. The strength of these assemblies was increased by repeating the threatening stimuli. These findings imply that even without a highly salient event, subgroups of neurons with different tuning specificity are loosely coupled probably by shared input and can be co-activated spontaneously. Recent studies have uncovered that spontaneously emerging assemblies reflect preconfigured dynamics possibly determined by inputs of the assembly-forming neurons[44,45]. Here, we demonstrate that spatially tuned and event-coupled cells are integrated into assemblies emerging before the aversive stimulus. When the salient event comes about, it reorganizes these assemblies by boosting the weight of members tuned for the event-defining stimulus. After the salient episode, remnants of these assemblies are still detectable and may be stabilized during ensuing sleep epochs to become memory traces[46]. Thus, air puff-activated pyramidal cells may be pivotal for remodeling the spatial representation in the dorsal

hippocampus to accommodate the aversive episode for guiding future threat-avoiding actions.

## Methods

### Animal care and housing
Male 8–12-week old, vGAT-iRES-Cre (Jackson Laboratory, JAX:016962) and vGlut3-iRES-Cre (Jackson Laboratory, JAX:018147) mice were used in the present study. Animals, two to three mice per cage, were kept in temperature- and humidity-controlled housing facilities (22 °C and 50 ± 5%) with a standard 12-h light–dark cycle, with food and water available ad libitum. All experiments were approved by the Animal Care and Use Committee of the Institute of Experimental Medicine and the Committee for Scientific Ethics of Animal Research of the National Food Chain Safety Office under the project number PE/EA/200-2/2020 and were performed according to the 2010/63/EU Directive of the EC Council. All efforts were made to minimize pain and suffering and to reduce the number of animals used.

### Surgical procedures
Surgeries were performed under general anesthesia (isoflurane 0.5-1.5%), and analgesic (Buprenorphine 0.1 µg/g, s.c.) was applied at the beginning of surgery. A small lightweight head plate was attached to the skull using Optibond adhesive (Kerr, Brea, CA, USA) and Paladur dental acrylic (Kulzer, Hanau, Germany). During multichannel recordings, mice were head-restrained with a downward tilted head position (pitch angle: 20 °). Cranial windows (1.5 ×1.5 mm) were drilled above the left and right hippocampus (AP, −2.5 mm; ML, +/− 2.5 mm) under stereotaxic guidance. For the ground electrode, a hole was drilled above the cerebellum (AP, −6,0 mm; ML, 2.0 mm). The craniotomies and drill hole were covered with fast sealant (Body Double, Smooth-On, Easton, PA, USA). After surgery, the mice were continuously monitored until recovered, then they were returned to their home cages for at least 48 h before starting habituation to the head restraint. Running wheels were added to the cages a few days before the headpost surgery.

### Behavioral training
Head-fixed animals were running on the perimeter of a 70 cm diameter disc. The self-rotating disc consisted of a circular path made from loop side of Velcro material with 190 cm running path, decorated with a variety of textures, shapes and colors allowing the mice to distinguish path segments (Fig. 1a, Suppl. Figure 5). The disc was self-propelled, and reward (RW) was delivered through a custom-made lick port controlled by a solenoid valve (The Lee Company, Westbrook, CT, USA). The animal's position was tracked using a rotary encoder (Pewatron, Angst+Pfister AG, Zurich, Switzerland).

After 3–5 days of recovery from headpost surgery, mice were placed on water restriction receiving 1–1.5 ml water per day. 3–4 days after water scheduling, animals were placed on the head fixation apparatus on a training rig for 1060 min sessions per day for 2–4 weeks. Mice were trained to run for reward. Reward was delivered if the mouse moved and the distance between reward deliveries was gradually increased in parallel with the improvement of the mouse's performance. If the mice ran more than -50 cm then the rate of reward delivery was reduced to once / rotation of the disc, and one place was designated as the reward site. Mice started to run, once they got the reward just at the reward site. During the first 2 weeks of training, mice had more (1–3) training sessions per day. Moreover, animals were supplemented with additional water after training sessions if they had consumed less than 1 ml/day. Mice were also trained on the recording setup for at least 2–3 sessions before recording. The belt the mice used for training was the same belt used during the recording session.

### Multichannel electrophysiological recordings
On the day of recording Buzsaki64 (Neuronexus, Ann-Arbor, MI, USA) or UCLA128 channel silicon probes (Masmanidis lab, UCLA 128 J, Los

Angeles, CA, USA) were lowered through the cranial window to the left and right dorsal hippocampus under isoflurane anesthesia (0,75–1,5%). Probes were coated with lipophilic fluorescent dye, DiI (Thermo Fisher Scientific, Waltham, MA, USA) for later histological verification of the location. Ground electrode was placed above the cerebellum. Mice were allowed to recover from anesthesia for -1 h before recording.

Head restrained mice were free to run, walk or sit on an air supported, free-floating, 20 cm diameter polystyrene ball. The probes in the hippocampus were advanced using a micromanipulator (David Kopf Instruments, Tujunga, CA, USA) until reaching the CA1 pyramidal layer, identified by increased occurrence of unit activity and the appearance of ripple oscillations. Recording was commenced after an approximately 1 h waiting period for letting the tissue settle around the probes. Electrophysiological recordings were performed by a signal multiplexing head-stage (RHD 128, Intan Technologies, Los Angeles, CA, USA) and an Open Ephys data acquisition board (open-ephys.org). Signals were acquired at 20 kHz sample/s (data acquisition software: Open Ephys 0.4.4.1). Recording started with a control, unstimulated session on the air-supported polystyrene ball, if not stated otherwise. Mouse locomotor activity on the ball was monitored with an optical computer mouse positioned close to the polystyrene ball at the equator. After 20-30 min recording on the ball, the context was switched to the cue-decorated disc, where the mouse started to run for reward and the recording with air puff, RW and other stimuli commenced. Air puff and TS locations were different from the RW locations on the disc. After the mouse had completed enough number of laps, we put him back on the polystyrene ball.

In experiments aimed to test the effect of the amount of sensory information on the air puff-evoked response, variable number of air puffs (6-12) were delivered pseudo-randomly with a minimum inter-stimulus time interval of 30 s on the sphere. During the following disc session, the air puffs were delivered the same way as described above. In the final phase of these experiments air puffs were delivered pseudo-randomly again on the sphere. To exclude that the order of contexts (i.e. ball-disc-ball) influences response characteristics, we changed the order of sessions in another set of experiments (disc – ball – disc). In all of these sessions, a novel disc was used in the second disc session.

In a further subset of experiments, following 2 air puff epochs on the disc, mild tail shocks were delivered at 2 locations non-overlapping with the air puff locations.

In experiments for testing the effect of neutral stimuli, tone stimulation was applied at random, nonoverlapping locations following the air puff epochs. Details of sessions can be found in Table 1.

The disc was fixed to a sliding platform which allowed the rapid switch between the disc and an air-supported polystyrene ball underneath the animal during recording (Supplementary Figure 5).

Custom-built microprocessor-based (Arduino) behavioral control system enabled the location-dependent delivery of reward, air puff (30 psi, 200 ms), tail shock (2*2 ms, 10 Hz, 1 mA) and tone (7.5 kHz, 200 ms) stimuli. Speed was calculated by a custom-written procedure in IgorPro. At the end of the recording, mice were transcardially perfused with 4% paraformaldehyde and the brain was removed for *post hoc* immunohistochemistry.

### Histology
For reconstruction of silicon probe positions, coronal or parasagittal sections (60 µm) were cut using a vibratome (Leica Microsystems, Wetzlar, Germany), mounted on microscopic slides and covered with mounting medium (Vectashield, Vector Labs, Burlingame, CA, USA). Silicon probe tracks were localized by imaging red fluorescent DiI that had been applied on the electrodes before implantation. Fluorescence signals (DiI) were inspected using an Axioplan 2 microscope (Carl Zeiss, Oberkochen, Germany). Images were aligned with the corresponding sections of the stereotaxic mouse brain atlas to enable reconstruction of the trajectory.

## Data Analysis

All in vivo data were analyzed in Igor Pro 8 or 9 (Wavemetrics, Lake Oswego, OR, USA). If additional software was used, it is stated at the respective analysis.

## Spike sorting and neuron classification

Neuronal spikes were detected and automatically sorted from the high pass filtered (0.3 – 6 kHz) recordings by a template matching algorithm using the Spyking Circus software[47], followed by manual curation of the clusters using the Phy software[48] to obtain well-isolated single units. Multiunit or noise clusters were discarded from the analysis. Spike sorting quality was assessed with refractory period violation, and visual inspection of auto- and cross-correlations; poor-quality clusters were discarded. A burst index was computed by calculating the ratio of the average values in 3-20 ms and 20-100 ms windows of the single units' autocorrelograms. Putative interneurons were defined if the spike width at the half maximum was less than 0.3 ms and the burst index was less than 2. Putative pyramidal units were defined with spike width at the half maximum more than 0.3 ms and burst index greater than 2. All other single units outside these categories were not included in the analysis.

Superficial-deep location of the putative pyramidal units were defined by their soma (maximal spike amplitude) distance from the ripple maximum.

## Detection of significant air puff response

On the average peristimulus histogram of each unit a baseline segment between −4 – 0 s preceding the stimulus onset was defined. The z-scored peristimulus histograms were calculated by using the corresponding baseline mean and SD values. Air puff-activated pyramidal cells (aAP-Pyr) were defined as putative pyramidal cells with responses exceeding 2 z-score values[49,50] for at least 50 ms duration in a 3 s long time window following air puff and these responses could be detected at 2 or 3 out of 3 or 4 air puff-locations. The first 20 ms bin where the z-scored air puff response elevated above 1 z-score values were detected. In this starting bin the average latency of the first spike from stimulus onset was computed for all air puff stimuli. AP-evoked suppression was defined as drop of activity below z-score of < −1 postair puff.

## Nomenclature of neuron groups based on their response to air puff and cell type

Terms were composed of two parts: first part indicates the type of response:

aAP; activated by air puff;
iAP: inhibited by air puff;
nAP: nonaffected by air puff;
The second part of the term corresponded to the cell type:
Pyr: pyramidal cells composed of two subgroups:
PC: place cells
nPC: non-place cells;
IN: interneurons.

For example, aAP-Pyr corresponds to air puff-activated pyramidal cells.

## Place cell analysis

The 190 cm long running path of the disc was divided into 2 cm long spatial bins. A place field was defined as an at least 3 bin long (6 cm) continuous segment of the non-smoothed tuning curve where the firing rate elevated above 5 % of the peak firing rate of that unit. The place field start and end location were detected as the threshold (5 % of the peak firing rate) crossing locations. The place field detection was repeated for each unit for identifying occasional multiple place fields. For place field stability criteria each no stimulation epoch (preceding the 1., 2. and 3. air puff epochs) was divided into two halves and the Pearson's correlation coefficient (R) of the non-smoothed tuning curves of the two halves were computed. Only place fields with significant correlations and at least 0.3 R value were accepted. Putative pyramidal units which fulfilled the above criteria were regarded as place cells. Spatial information was calculated as described previously:[24]

$$Spatial\ Information = \sum_{i=1}^{N} p_i \frac{F_i}{F} \log_2 \frac{F_i}{F}, \tag{1}$$

$i = 1,…, N$: spatial bins; $p_i$: probability of occupying the $i^{th}$ bin; $F_i$: mean firing rate of the neuron in the $i^{th}$ bin; $F$: overall mean firing rate of the neuron.

## Identification of monosynaptic connections

To detect monosynaptic connections between putative pyramidal cells and interneurons the methods described earlier was used[51,52]. Raw cross-correlations (1 ms binning) between putative pyramidal cells and interneurons were calculated. Predictor cross-correlations were computed by convolving the raw cross-correlations with a partially hollow (hollow fraction 60%) Gaussian kernel (Supplementary Fig. 4c) smoothing out sharp cross-correlation features. Poisson distribution was used to estimate the probability to obtain the values in the raw cross-correlations given the corresponding predictor cross-correlation values[51]. A significant monosynaptic connection between a putative pyramidal cell and interneuron was defined if the peak of the raw cross-correlation was in 0–3 ms time window and the probability of the peak value was $P < 0.001$. Zero-lag peaks were excluded. The average number of monosynaptic pyramidal-interneuron connections was determined in different categories of presynaptic putative pyramidal neurons and postsynaptic interneurons.

## Ripple detection and ripple coupled firing

Hippocampal pyramidal layer LFP was band pass filtered (120–250 Hz) and the Hilbert magnitude was computed. Ripple events were detected during immobile periods and were defined as epochs continuously larger than mean + 1 standard deviation and the peak ripple magnitude was greater than mean + 3 standard deviations. Events shorter than 20 ms were discarded. Ripple participation of a unit or assembly was calculated as the ratio of the number of ripples with activity of the unit or assembly divided by the total number of ripples in that session. To further characterize the ripple-related firing activity the ratio between the total spike number during ripples and the number of spikes during immobile state was determined. Additionally, the average spike number during active ripples (in which the actual unit is participating) were computed for each putative pyramidal unit.

## Theta phase and phase precession analysis

Theta phase was determined by applying Hilbert transformation on the bandpass filtered (5-12 Hz) CA1 pyramidal layer LFP. In case of both pyramidal cells and interneurons instantaneous theta phases were assigned to the spikes. Based on the spike phases phase preference histograms were created for interneurons with a bin size of 20°. For place cells to visualize theta phase precession spike phase was plotted against the location on the disc. Circular-linear regression was calculated as in[53]. Briefly, the slope parameter of the circular-linear regression was determined by maximizing the mean resultant vector length. The phase offset (start phase) was calculated using the slope parameter as in Kempter et al 2012. The end phase was determined by assuming a linear model between the spike phase and location in defined place field range. Similar analysis was performed in the case of air puff-activated pyramidal cells but linear regression was applied between theta phases and the corresponding air puff response time window.

## Assembly analysis

Neuronal assembly detection analysis was based on independent component analysis of the neuronal covariance matrix[30]. The spiking activity of putative principal neurons were counted in 25 ms bins followed by z-score normalization. Principal component analysis was performed on the correlation matrix of the pyramidal neurons. The number of significant principal components were determined by using a threshold value derived from Marcenko-Pastur distribution. An independent component analysis extracted the assembly patterns from the significant principal components. See Supplementary Figure 10 for the illustration of the procedure.

To calculate assembly reactivation strength over time the outer product of each assembly's weight vector and the population vector in the 25 ms time bins was calculated representing the similarity between each assembly and the population firing patterns in consecutive time bins. Assembly reactivations were defined as peaks above a threshold of 5[54].

To detect air puff assemblies, synthetic air puff patterns were created in each session (Supplementary Figure 10). In this synthetic air puff pattern, each neuron's weight was set to zero except that of aAP-Pyr cells. The weights of these cells were set so as the sum of squares of weights gave 1. A similarity index (absolute value of the inner product of the weight vectors and the synthetic air puff pattern[55]), was computed between the synthetic air puff pattern and each of the assemblies in that session. Significant similarity between the synthetic air puff pattern and a "real" assembly was defined using bootstrapping method (Supplementary Figure 10). A similarity index distribution was created for each comparison by iteratively (1000 times) mixing the unit weight values in the real assembly and computing the similarity index in every iteration. Similarity between assemblies was significant if the similarity index value is located outside the 1 % of the bootstrapped distribution.

Assembly detection was repeated after removing air puff epochs from sessions. In these calculations the spikes during laps containing air puffs were deleted before the assembly detection procedure.

For assembly analysis both Python and IgorPro 9 scripts were used.

## Bayesian decoding

Probability-based decoding of animal location on the disc was performed using the Bayesian algorithm as described previously[31], assuming that spikes have Poisson distributions and place cells are statistically independent. Periods when the running speed was above 2 cm/s for at least 0.5 s were analyzed and theta cycles were used to estimate the mice's positions. Position reconstruction error in each theta cycle was calculated as the difference between the decoded position and the animal's actual location on the disc. To investigate the alteration of reconstruction error upon air puff application we divided the preceding no-stimulation epochs in two halves. Control reconstruction error was calculated by decoding the second half of the no-stimulation epoch based on the first half. Then it was compared to the decoding error calculated by decoding the air puff epoch locations based on the second no-stimulation half. Average decoding errors were computed in 2 cm long bins.

## Decoding of reward location

In each investigated session the position distribution of the normalized occurrence of the decoded peri-reward (± 20 cm) location was computed for the 1st and 2nd no-stimulation epochs. A similar distribution was created for the 1st and 2nd air puff epochs. To represent the decoding of other locations, random positions were selected 1000 times different from the peri-reward and the animal's actual positions and position distributions were computed again.

To reveal the cause of the systematic shift of estimated position to the reward zone during air puff delivery, we analyzed the place field of pyramidal cells differently contributing to position estimation. Theta cycles coding reward locations (reward coding theta cycles) were detected in a 3 s time windows following air puff onsets in the first air puff epoch. Then, the average firing activity of individual putative pyramidal cells in these reward-coding theta cycles were computed. Based on the firing rate within reward-coding theta cycles, place cells were sorted and divided into firing rate quintiles i.e. from the least to the most active pyramidal cells Finally, the z-score-normalized tuning curves of pyramidal cells in each quintile were averaged resulting a normalized mean tuning curve for every firing rate quintile (upper 80-100%, 60-80%, 40-60%, 20-40%). Difference between the normalized average tuning curves in the fifth and fourth quintiles (80-100% and 60-80%) were calculated and the point-by-point statistics were computed (Fig. 9 and Supplementary Fig. 11, Wicoxon signed rank test) along with per-quintile tuning curves and their averages.

## Quantification and statistical analysis

All statistical analyses were performed with standard IgorPro9 functions. Wherever possible, raw data points are presented overlaid on box and whisker or violin plots representing median, interquartile (25−75%) and 10−90% range. In specified cases, mean and standard deviation were provided. Standard error was used only for illustration purposes, indicated in the respective figure legends. For unpaired two-sample comparisons, the parametric Student's t-test (two-sided) or Wilcoxon-Mann-Whitney test was used. For paired comparison, the nonparametric Wilcoxon signed-rank test was utilized. For the majority of multiple comparisons, Kruskal Wallis test with Dunn-Holland-Wolfe post hoc test was applied whereas for normally distributed variables ANOVA with Tukey's honest significant difference or Dunnett's post hoc test was used. For the comparison of two variables measured in consecutive time points, 2-way repeated measures (r.m.) ANOVA was used followed by Tukey's test. Two-way r.m. ANOVA was carried out using Microsoft Excel (Microsoft Corp., Seattle, WA, USA, Real Statistics Resource Pack by Charles Zaiontz). A comparison of distributions was carried out by Kolmogorov-Smirnov test. All figure legends include information about the statistics used with the number of cases, value of the statistics and P. In the case of parametric statistics, normality was tested using the Shapiro-Wilk test, and equality of variances was tested using the F test or Levene's test. For significance, $*p < 0.05$ and $**p < 0.01$ was used.

## Reporting summary

Further information on research design is available in the Nature Portfolio Reporting Summary linked to this article.

# Data availability

The electrophysiological recordings generated in this study have been deposited in the Zenodo repository under a Creative Commons Attribution 4.0 International Public License (CC BY 4.0). Accession codes: https://doi.org/10.5281/zenodo.8339978, https://doi.org/10.5281/zenodo.8341204, https://doi.org/10.5281/zenodo.8341254, https://doi.org/10.5281/zenodo.8341258, https://doi.org/10.5281/zenodo.8341264, https://doi.org/10.5281/zenodo.8341268, https://doi.org/10.5281/zenodo.8341280, https://doi.org/10.5281/zenodo.8343580, https://doi.org/10.5281/zenodo.8343582, https://doi.org/10.5281/zenodo.8343572, https://doi.org/10.5281/zenodo.8343509, https://doi.org/10.5281/zenodo.8343558. A Source Data Table is also provided with this paper. Source data are provided with this paper.

# Code availability

The Igor Pro scripts used for the analysis of data are available at zenodo.org under a Creative Commons Attribution 4.0 International Public License (CC BY 4.0): https://doi.org/10.5281/zenodo.8343626.

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

## Acknowledgements

We thank Emőke Szépné Simon for essential technical assistance and Kathrin Petrik for her assistance with the present work. We thank Scidraw.io for providing the availability of the drawings doi.org/10.5281/zenodo.3925913 and 3925935. We are grateful to Drs Sanja Bauer Mikulovic, Balazs Hangya and Prof. Gyorgy Buzsaki for critically reading the manuscript. This work was funded by National Research, Development and Innovation Office, Hungary FK129019 grant to A.M.B. and K132735 grant to V.V., Eötvös Loránd Resarch Network Distinguished Reserach Project grant to V.V., the NRDI Office of Hungary within the framework of the Artificial Intelligence National Laboratory Program (RRF-2.3.1-21-2022-00004) and within the framework of the Translational Neuroscience National Laboratory (RRF2.3.1-21-2022-00011) to V.V. A.M.B. was supported by the New National Excellence Program of the Ministry for Innovation and Technology (ÚNKP-18-5, ÚNKP-19-4 and ÚNKP-20-5) and the Bolyai János Research Fellowship of the Hungarian Academy of Sciences.

## Author contributions

A.M.B., M.J and V.V conceptualized the study. A.M.B. designed the experimental setup. M.J, A.M.B. and F.V.N. performed the experiments. A.M.B. and M.J. analysed the data. M.J., A.M.B. and V.V. prepared the manuscript. A.M.B. and V.V. acquired funding. V. V. supervised the study.

## Competing interests

The authors declare no competing interests.
