## [Peer Review File · Nature Communications]

Aversive stimulus-tuned responses in the CA1 of the dorsal hippocampus.REVIEWER COMMENTS

Reviewer #1 (Remarks to the Author):

The manuscript by Barth and colleagues 'Aversive stimulus-tuned patterns in the CA1 of the dorsal hippocampus' addresses hippocampal cells responsive to mild aversive stimuli per se irrespective of location (or episode segment) of occurrence. The authors recorded neuronal spiking in the dorsal CA1 area as the animals were running on a 'contextualized' (that is, richly equipped with various distinctive tactile cues) horizontal wheel and were stimulated by air puffs at various locations (in trial blocks) interleaved with blocks of control trials (no aversive stimuli), and found that a good proportion of cells responded to such stimuli whatever the location. The authors discovered a coordinated subpopulation of pyramidal cells (largely different from place cells) and interneurons in dorsal CA1 that selectively respond to (and as the authors put, 'encode') mild aversive stimuli.

I was very enthusiastic reading the manuscript. The main finding is novel and significantly advances our understanding of hippocampal function, and together with an analogous study on somewhat similar effects of rewarding stimuli (Gauthier et al. 2018 *Neuron*, discussed in detailed in the manuscript) puts a new angle on the nature of contributions of the dorsal hippocampus (in particular the CA1 area) to information processing and associational learning in rodents. The data presented support the main conclusion of the study, the experiments are well designed and carefully executed. Therefore, I suggest that the results of this study are of interest to a broad neuroscience community. However, some important details are not sufficiently elaborated, and some conclusions are not unequivocally supported by the data presented. Also, in some instances the description of the results and analysis methods is not sufficient to evaluate (and certainly not to repeat) the conclusion drawn from the experiments. Below I list some general and some more specific criticisms, comments and suggestions, which I believe may help the authors to improve their manuscript.

General comments

1. The authors collect and present convincing evidence that the cells responding to air puffs do so to the stimulus 'per se', and that these responses are different from place fields and are not simple place-stimulus associations. However, they fail to address the role of associational learning and predictability in the development of the responses. Although the authors allude to the instantaneous occurrence of the response, a systematic analysis of first ever responses and the development of responses across trials, blocks and sessions is missing. In the example presented in Fig. 1. the first 3-4 trials of the first 'stimulus block' seem to lack responses, which then quickly develop in the upcoming trials. Is this a first stimulus session in the life of this animal? If yes, is this a systematic variation of responses? A decrease in responses occurs on the spherical ball compared to the wheel, which according to the authors interpretation is a consequence a 'non-spatial' context. Although this experiment is not properly described in methods (most importantly: were the stimuli applied regularly or randomly on the ball?), the predictability of these responses certainly decreases on the ball, which offers a plausible alternative explanation. The animals can certainly predict the air puffs as indicated by their slowing down before (perhaps in expectation of) it (Fig. S1c). One may wonder how responses to air puffs may appear when

the stimuli were truly random (unpredictable) within the same context. (As a side note: for historical reasons we often refer to self-propelled repeating episodes as spatial, from the animals' perspective there is arguably very little 'spatial' in moving a structured horizontal wheel by his limbs; see Aronov et al. 2017 Nature or Pastalkova et al., 2008 Science).

2. The authors present no convincing data to support their claim that the 'AP-cells' and the place cells represent separate, distinct populations. The proportion of place cells and air-puff responsive cells in pyramidal cells are 53% and 40%. Assuming independent distributions one may expect 21% of pyramidal cells to respond to air puffs and be a place cell, pretty close to the observed 26 %. This suggests that amongst pyramidal cells having a place field and responding to air puffs combine independently. The firing rate differences between air puff responsive and non-responsive cells are not evidence in this respect (distributions that may indicate clear separability of populations are not reported). Also, the anatomical segregation of the two populations is not demonstrated unequivocally (see my comments below). Therefore, to refer to these cells as a sub-circuit seems far-fetched and a bit misleading (even in spite of the results of assembly analysis).

3. I suggest the authors revise the terminology they use/introduce in this manuscript. By coining the term AP-cells the authors suggest that these cells 'evolved' to encode air puffs. While air puffs are very useful to apply mild aversive stimulation in a quantitative manner, their occurrence and relevance in natural habitats is not clear, and it is hard to imagine there are cells selective for this particular modality of aversive stimuli. It is clear these cells do not respond to painful tail shocks and neutral sensory stimuli. It would be nice to see if the response generalizes to stimuli of similar valence (mild aversive) but different modality. In addition, the abbreviation AP is extensively used as for action potential in neurophysiology, which may confuse readers. The authors use AP-responsive cells inconsistently. In general air puff response may be inhibition or excitation. The data presented shows that the air puff induces a rather complex response on the population level with some cells (in fact the majority) decreasing and others increasing their firing. There is no scientific justification why increases are more relevant than decreases. Finally, the term conjunctive AP-cells suggests that these cells respond to position and air puffs in a combinatorial way, like conjunctive HD-grid cells in the mEC. The authors do show interactions between responses, yet the response properties of these cells are not conjunctive in the strict sense.

It is more useful to report standard deviation than standard error as a data dispersion measure. Why did the authors chose otherwise?

Specific comments:

Abstract, line 22-23: Rephrase and separate this sentence.

Abstract, line 25: The sentence implicates that place cells are responsive to non-salient stimuli, but the tone stimulation did not evoke cellular responses at all.

p3, line 93; Please report statistics for the proportion of cells preserving their air-puff associated firing characteristics (negative or positive) over numbers of different air puff locations (e.g. 2/3, 3/3, 3/4 or 4/4).

p3, line 106 There is a discrepancy between the text (reporting proportion of AP cells being not place cells) and the total numbers (given as all pyramidal cells). The reporting of data needs to be consistent throughout the manuscript.

p3. The spatial distribution of different cell categories along the radial axis in the pyramidal layer is insufficiently described in the manuscript. The mean distance from the center of stratum pyramidale being around zero may mean uniform distribution along the radial axis as well as enrichment in the middle (as suggested by the authors). In addition giving the SEM and not SD as dispersion measure, prevents the reader from inferring or even comparing the distributions. In this case also the ANOVA is inappropriate as the relevant question is whether the distributions are different and not if one group is on average more superficial or deeper than the other. I suggest to include the actual distributions of distances.

p4, line 143; Please provide statistical evaluation of this data considering that perhaps the number of interneurons firing on the trough and on the peak are also much different. Defining interneurons firing on the trough and on the peak is not trivial (there are other interneuron types firing on intermediate phases, and the within-group variability is also high). Please provide the details.

p4, line 143; Since the number of potential presynaptic and postsynaptic partners varies significantly across the different groups (multiplication further increases the differences in the numbers of potential contacts) comparing the numbers of detected functional interactions is not necessarily reflecting actual differences. Please use likelihood calculus.

p5, The last sentence of the § in lines 185 186 is incomprehensible.

p5, line 201; 'use not associated with' or 'not coupled to' instead of 'uncoupled'.

p5, line 223; 'give us a clue' instead of 'give as a clue'

p5, line 226-8; The authors claim that the phase precession in AP cells started at later and ended at earlier phases than in place cells. One may arrive to a conclusion that this expands the phase range covered during the phase precession, which is the opposite of what the authors actually report. In fact the interpretation of 'earlier' and 'later' is problematic with circular data. For example the start of the phase precession in AP cells is just after the trough in AP cells (29°) and just before the peak in place cells (147°), which is almost opposite phase of the cycle (which is earlier or later?). In contrast in the four representative examples of Fig 5, the difference in the starting phases in AP and place cells appears to be minimal. Please display the individual ranges in Fig 5, as these are not trivial from the start and end distributions (and is as informative as that). Furthermore the description of methods for determining start and end phases of phase precession is insufficient, and it is far from trivial how one derives these from the circular-linear regression (also the steepness of the position-phase relationship may not be stationary across the entire place field).

p6; In the text (line 248-9) the authors claim that correlations within control place cell maps are higher than correlations between the maps recorded during stimulation and control including the pre vs. Post control sessions as well (text in brackets), but the legend of the referred Fig 6c indicates that it was pre-pre, ap-ap and post-post but not pre-post correlations compared to ap-pre and ap-post. Please clarify, and if pre-post correlations are not calculated, include them as well. A detailed description of place map

calculus and how the above correlations were calculated is also missing from the methods section (if there was any smoothing applied, for example).

p6; Ripple participation analysis should account for any variations in the baseline firing rate. The current calculus will not adequately measure specific ripple participation but will instead depend on other variables as well.

p6; The terms assemblies and patterns are used as synonyms. This is confusing, the two terms aren't interchangeable.

p7 line 290; 'not locked to any location...' instead of 'unlocked to any location...'

p8, line 347: '...to the aversive stimulus, thus when...' instead of '...to the aversive stimulus thus, when...'

p9, line 380: Dorso-ventral axis of the hippocampus is different from the deep-superficial (radial axis) which makes this sentence easy to misunderstand.

p10, line 442: '...resembling the threat-related ones...' instead of '...resembling to these threat-related patterns'

p21 methods, line 886 onwards; Please indicate if the z-scoring was performed over averages or binned firing rate changes and over what extent (length) and type (brain state, movement) of recording. Also, the reporting of firing rate changes is often not clear (what was averaged, peak or mean rates, etc.), and obscure abbreviations are used on axes (Norm. Sp. No. *10⁻³).

p22 methods, line 906; There are several well established methods in the literature to evaluate potential monosynaptic connections. The method used by the authors appears somewhat arbitrary, please provide the references and the rationale.

Fig 1a: The arrow indicating the rotation direction or the mice's nose should point the other way.

Fig 1c and S1a: The purple lines in the histograms are easily confused for a histogram data point. Please replace by a more distinct sign (an arrow for example); Peri AP firing histograms are not interpretable in control trials (where there is no air puff).

Fig. S1e: I could not find the colour scale bar.

Fig 1h: I don't understand what is on the x axis. Is it the log₁₀ of the latency in s?

Fig 2b: The violet bars are difficult to discriminate from the colour-scaled image behind.

Fig. 6a: The label and units for x-axis is missing.

Fig. 6c: The colours of the control and AP distributions are hard to distinguish. Please change.

Reviewer #2 (Remarks to the Author):

Barth et al. adopt state-of-the-art behavioral and recording techniques to investigate if there is a subcircuit specialized for aversive stimuli in the hippocampus. The authors elucidate a cell population comprising both pyramidal cells and interneurons that selectively fire for an aversive air puff response. The firing of these cells is further modified depending on the availability of spatial information in the environment and the salience of the aversive stimuli. The authors then impressively show that these cells exhibit theta phase precession and participate in ripples, suggesting some kind of sequential coordination between these aversion-responsive cells that may guide behavior and memory consolidation. Lastly, the authors suggest that aversion is represented by pre-formed cell assemblies before even receiving an aversive stimulus. The authors also show through a decoding analysis that following a shock, the hippocampus increases its representation of the reward zones.

On a general level, the paper is interesting but my enthusiasm is muted by some unsupported claims, as well as somewhat careless and/or inconsistent presentation of abbreviations, statistics, and presentation. Statistics are also not presented altogether in some cases.

Figure 4 somewhat implies that there is not a true aversion circuit in the hippocampus. And Figure 6 is interesting, but again suggests that aversion also influences non-AP place cells. This isn't too problematic, since it has been shown that spatial cells respond to all kinds of sensory stimuli. But it does bring into question how specific this "aversion circuit" really is for aversive stimuli.

Figure 5 is impressive and interesting, showing that theta phase precession exists even for non spatial coding.

The assembly analysis is interesting in concept, but (likely) suffers from such low sample size of available AP assemblies such that statistics cannot be run properly. The low number of AP assemblies (~7% of all detected assemblies) also does not reconcile the earlier results showing that a pretty significant proportion of ALL hippocampal cells (>30%) can encode an air puff and/or a tail shock. It seems that the authors are making a huge stretch in their interpretation here and I would advice caution.

While the positional decoding analysis is interesting in terms of implications, those results do not tie together the main focus of the narrative presented through Figures 1-7 which were focused on aversion-responsive cells. The lack of a circuit-level explanation for why the decoding results are the way they are renders the last figure more questionable than insightful. I suggest that the authors try to search for a physiological explanation of the decoding analysis by further analyzing the results presented in Figure 6. Otherwise, the findings of this decoding analysis under shadow the significance of all the other figures because the decoding analysis counterintuitively suggests that this "aversion circuit" is not as important as the authors think.

Figures 7 and 8 are the most problematic for me. In many ways, it feels like the authors are overly ambitious in trying to link their prior results throughout Figures 1-6 in a greater context. As such, both figures appear disconnected from the remaining narrative.

Major Comments:

Figure 1:

- No supplementary material is shown, making it impossible to review claims that refer to them.
- Figure 1f: There are 16% of cells that decrease activity and are place cells. By definition, place cells should be spatially selective and silent when the animal is not in the place field. Since the air puff was administered at a minimum of 3 different spatial locations to ensure non-overlap with the place field, it is curious that a place cell could suppress their (already silent) firing even more following an air puff. Why is this the case? And as an aside, it would help to show individual cell examples of this kind of cell, as well as other kinds of AP cells (those that may have overlapping place fields or cells that are not as clean in response) in the supplementary.
- Figure 1j: This is hard to reconcile given that in Figure 1f, there are more AP cells that reduce their firing following an air puff (24%) as opposed to those that increase their firing following an air puff (16%). On another note, it is not clear what cell population is being shown in this panel. Is it all AP cells? Also, the figure caption indicates that a 2-way ANOVA was used. But the panel only groups the data by location (pre, AP and post). The second factor of “stimulus” is not shown. This is confusing because it seems a 1-way ANOVA could’ve sufficed for this panel.
- The two panels in Figure 1k should be combined into a 2-way repeated measures ANOVA, the repeated factor being movement (immobile or moving) and non-repeated factor being cell type (AP or non-AP). Then, follow with a significant interaction effect, appropriate repeated measures correction and post hoc tests. If a repeated measures design is not appropriate for whatever reason (such as missing data), then the stat design should be at the very least be a regular 2-way ANOVA followed by a significant interaction effect and appropriate post hoc tests.
- As it currently stands, given the nonsignificant result, it is inappropriate to state in the text that a certain population of pyramidal cells is more superficial than another.

Figure 2:

- Figure 2b: Second interneuron example is not convincing. Is there a better example?
- Figure 2f: Are biphasic interneurons consistently biphasic? I worry that the biphasic activity is just spike contamination from the interneuron’s natural high firing rates that mixes with suppressed activity following an air puff.
- Figures 2i-k: Doesn’t a monosynaptic connection imply that the interneuron could be synapsing onto the pyramidal cell as well?
- Despite no significance in Figure 2j, the authors make an unsupported claim: “AP-cells were more prevalent among the presynaptic partners of AP than of non-reacting interneurons”.

Figure 3:

- Statistics are not reported for Figures 3c and 3g.
- Lines 173-174: “Thus, AP-triggered interneurons tended to be modified in the opposite direction by the cue rich to nonspatial context switch.” This statement is inaccurate since facilitation and suppression are BOTH enhanced by the air puff in pyramidal cells. There is no rationale here to suggest a story in which the interneuron population is affected in the “opposite” direction since there is no opposite direction to begin with.

Figure 4:

- Figure 4 brings into question whether there is really a hippocampal subcircuit for aversion. This figure shows that air puff cells for the most part are unrelated to tail-shock cells. In other words, AP cells don't serve a general role for aversion (or at least, only a very small percentage do – 8%).
- The counterargument is that these results is sort of akin to global remapping of place cells between contexts. Some place cells are silent in one environment, whereas they are active in another. Perhaps this is what is taking place here in the context of aversion. However, it seems a little unlikely to me, given that the spatial context itself is unchanging. (And, we saw from Figure 3 how much influence the spatial context exerts on AP cells.) And furthermore, the fact that you then have another subset of cells that is specifically responsive only to the reward makes it more likely that these “AP cells” are not really part of an aversion subcircuit but serve a broad hippocampal function to store the animal's experience, or salient stimuli in general.

Figure 5:

- Figure 5: Do conjunctive AP cells have the same phase precession range for both the AP stimulus and a normal place field? Or is the phase range greater for the place field only?
- Figure 5: There is no way to determine if recruitment of conjunctive AP cells during ripples is for consolidation of the AP (aversion) or place fields (space). It is recommended to run this analysis for pure AP cells and see if they participate more during ripples than non-AP cells.

Figure 6:

- Line 242: “Overall, in-field firing of place cells was suppressed if the AP was co-located with the place field whereas their activity”. This statement is not entirely correct given that about 10-15% of place cells seem to increase their firing following an air puff. I would suggest modifying the sentence to make it more conservative.
- Continuing off the same point, could it be possible that some cells had an air puff towards the end of the cell's place field? In such a case, the suppression of activity is natural, and not due to the AP.
- Line 243 and Figure 6b: First is not clear why a z-score less than 2 is nonsignificant. No statistical explanation is given. Second, if one were to run a statistical test, it seems to me that the best approach here is to compare pre-AP response versus post-AP response using a repeated measures test like the sign rank test.

- No stats are provided for the cumulative distribution graph in Figure 6c. Also, please use different colors/line markers to differentiate different comparisons. There is no way to differentiate which line is which comparison.
- Difficult to assess the merits of Figure 6 without access to the supplementary figures.

Figure 7:

- Can the authors provide a layman explanation of the assembly analysis for readers that are unaware of the paper that the analysis is based on?
- Figure 7a: This figure looks more like an example of a single cell rather than an assembly.
- Figure 7b: How can an assembly really be an assembly if the population is 2 or less?
- Figure 7e: A two-way ANOVA should be used here. Also, in the figure caption, please provide description statistics.
- Lines 296-299: “We were able to identify pre- and post-session ensembles resembling the AP-patterns but the weight of AP-cells in these patterns was significantly lower in most of the cases than in similar AP-assemblies isolated during the AP session”. This is a false statement given that the panel shows non-significance.
- Figure 7g: No post-hoc tests are carried out between the purple and black bars, or between the purple bars. Since the argument in this figure is that assembly strength increases, statistical significance should be shown to support this claim.
- My general worry in this figure is aside from the fact that assemblies have such little number of cells, the actual number of AP assemblies is also extremely low (only 7 versus 84 non-AP assemblies). The low assembly population and count do not really reconcile the results presented in Figures 1, 2 and 4 where >30% of cells have some kind of AP response. It therefore seems like a stretch to say that representation of an aversive event is allocated to a pre-formed aversion assembly.

Figure 8:

- Figure 8c: Please make sure the y-axis scale bar ranges are the same and report descriptive statistics (such as mean and s.e.m. or std) for the decoded results.
- Figure 8 is potentially very interesting, but it falls short without an explanation for why the air puff reconfigures the place map to encode the reward areas more. For instance, Figure 8 suggests that this shift in place map has nothing to do with AP cells. Therefore, the question is: “what is the circuit-level mechanism that explains why the air puff is causing the change in decoded location”. If the conjunctive AP pyramidal cells aren’t responsible, then the answer must lie in the non-AP cell population. Logically, if there is an increase in decoding of reward location instantly following an air puff, then it suggests perhaps one of two explanations:
 1. The training data of each air puff epoch is “the second half of the preceding control epoch” (Line 322). Since these control epochs are interweaved between air puff epochs, it is possible that there was place cell remapping (as a result of air puff; Figure 6) during these control epochs. Is this true? Can it be shown that place cell remapping in Figure 6 was more biased towards the reward zones? For example, in Figure

6a, in the “outfield” graph, after the air puff it seems like there is an increase in spiking in many cells. Therefore, the increase in reward representation in place cells is due to increased remapping of non-AP place cells towards the reward zones.

2. An alternative reason for increased reward coding is that following an air puff, there is an increase in activity of place cells whose natural place fields are the reward zones. Again, in Figure 6a, the increased activity of some cells in the “outfield” graph might belong to those place cells with reward zone place fields.

- But also, regardless of which of the above scenarios is true, Figure 8 makes it unclear just how important this “AP cell/aversion subcircuit” is for reconfiguring the place map towards positive valence stimuli. If anything, the results argue that there isn’t a true subcircuit for aversion, but the place code in general has the capacity to reconfigure its responses in the face of aversive stimuli.
- Lastly, what if the reconfiguration is not biased towards reward or positive valence? The reward zones were the only other salient locations in the task. What if the reconfiguration is really biased towards any other non-air puff location in the environment that has some basic level of salience?
- Figure 8 decodes air puff epoch locations using the preceding interleaved control epoch. It is unclear what “changes” have occurred to neural activity during these control epochs as a result of preceding air puff epochs. Do the results presented in Figure 8 hold true if air puff trials are decoded using ONLY the pre-AP control epochs as training data? The pre-AP control without any air puff serves as the true baseline control.

Minor Comments:

- Figure references are not in order. For example, Figure 1c is referenced before Figure 1d on line 86. Or, when Figure 7f is referenced before Figure 7e. On a related note, sometimes figures are entirely unreferenced such as Fig. 1j.
- Figure 1: In regard to selection of AP cells from the methods: “AP cells were defined as putative pyramidal cells with responses exceeding 2 z-score values for at least 50 ms duration in a 3 s long time window following AP and these responses could be detected at 2 or 3 out of 3 or 4 AP-locations.” Why do AP cells not all respond at all 3 or 4 of the air puff locations?
- Figure 1: Please define “sp” in the figure caption.
- Figure 1 caption at line 467 uses the AP abbreviation before defining it.
- Figure 1 caption at line 476: “Percentages” should be written in place of “Ratios”.
- Figures 3b and 3f show proportions, but percentages are reported in the figure caption.
- Figure 4c shows percentages, but the caption refers to the data as “ratios” and the reported data are provided as raw counts. Please stick to one type of reporting data.
- Figure 4: Please define TS and RW in the figure caption before using these abbreviations.
- Figure 4: Were the shock, tone and reward administered in the same locations as the air puff?

- Figures 4b, 4e: For all axis labels, only the first word should be capitalized. Figure 4b x axis is “Peri event time”. But Figure 4e x axis is “Peri Stim Time”.
- Significance stars between figures are inconsistent. For example, Figure 3b uses a single star for very small p values. But then Figure 5 uses two stars for similar p values. Please adhere to a single reporting standard such as: *: $p < 0.05$, **: $p < 0.01$, ***: $p < 0.001$. And please indicate this reporting criteria in every figure caption. (I.e., at the end of each caption, write “For all panels, *: $p < 0.05$, **: $p < 0.01$, ***: $p < 0.001$ ”).
- Figure 5: In the panels, please indicate that phase unit is degree, and ripple participation is percentage.
- Figure 6c: Please indicate units for distance.
- Figure 8a: Spelling error: “Real positon”.

We would like to thank the comments, criticism and suggestions of our Reviewers whereby we could significantly improve our manuscript.

Reviewer #1 (Remarks to the Author):

The manuscript by Barth and colleagues 'Aversive stimulus-tuned patterns in the CA1 of the dorsal hippocampus' addresses hippocampal cells responsive to mild aversive stimuli per se irrespective of location (or episode segment) of occurrence. The authors recorded neuronal spiking in the dorsal CA1 area as the animals were running on a 'contextualized' (that is, richly equipped with various distinctive tactile cues) horizontal wheel and were stimulated by air puffs at various locations (in trial blocks) interleaved with blocks of control trials (no aversive stimuli), and found that a good proportion of cells responded to such stimuli whatever the location. The authors discovered a coordinated subpopulation of pyramidal cells (largely different from place cells) and interneurons in dorsal CA1 that selectively respond to (and as the authors put, 'encode') mild aversive stimuli.

I was very enthusiastic reading the manuscript. The main finding is novel and significantly advances our understanding of hippocampal function, and together with an analogous study on somewhat similar effects of rewarding stimuli (Gauthier et al. 2018 Neuron, discussed in detailed in the manuscript) puts a new angle on the nature of contributions of the dorsal hippocampus (in particular the CA1 area) to information processing and associational learning in rodents. The data presented support the main conclusion of the study, the experiments are well designed and carefully executed. Therefore, I suggest that the results of this study are of interest to a broad neuroscience community. However, some important details are not sufficiently elaborated, and some conclusions are not unequivocally supported by the data presented. Also, in some instances the description of the results and analysis methods is not sufficient to evaluate (and certainly not to repeat) the conclusion drawn from the experiments. Below I list some general and some more specific criticisms, comments and suggestions, which I believe may help the authors to improve their manuscript.

We would like to thank the Reviewer for the encouraging opinion, insightful and thought-provoking comments and questions highlighting both the strengths and weaknesses of our study.

General comments

1. The authors collect and present convincing evidence that the cells responding to air puffs do so to the stimulus 'per se', and that these responses are different from place fields and are not simple place-stimulus associations. However, they fail to address the role of associational learning and predictability in the development of the responses. Although the authors allude to the instantaneous occurrence of the response, a systematic analysis of first ever responses and the development of responses across trials, blocks and sessions is missing. In the example presented in Fig. 1. the first 3-4 trials of the first 'stimulus block' seem to lack responses, which then quickly develop in the upcoming trials. Is this a first stimulus session in the life of this animal? If yes, is this a systematic variation of responses?

We agree with the Reviewer about the possibility that association of the aversive stimulus and location may have modulated the response to air puff of activated pyramidal cells (formerly AP-cells in the previous version of the manuscript, now called aAP-Pyr, see our later remarks about nomenclature). To test this possibility, we compared the magnitude and duration of activation by the stimulus within air puff epochs.

Thus, the effect of the first, second, third etc. air puffs were compared within the 1st and 2nd air puff epochs. We could not find any systematic change of the air puff response, thus the response of air puff activated pyramidal cells is not modulated by an associative learning process. This analysis was inserted into the first section of the Results ("*Air puff activates a subset of both putative pyramidal cells and interneurons in the dorsal CA1*") and shown on Supplementary Figure 2. Notably, the stability of air puff-activated pyramidal cell response across air puff epochs (already described in the original version of the manuscript) and the lack of its tendentious change during repeated stimulation suggest that aAP-Pyr are tailored to reliably signal the occurrence of the aversive event to the hippocampal circuit.

A decrease in responses occurs on the spherical ball compared to the wheel, which according to the authors interpretation is a consequence a 'non-spatial' context. Although this experiment is not properly described in methods (most importantly: were the stimuli applied regularly or randomly on the ball?), the predictability of these responses certainly decreases on the ball, which offers a plausible alternative explanation. The animals can certainly predict the air puffs as indicated by their slowing down before (perhaps in expectation of) it (Fig. S1c). One may wonder how responses to air puffs may appear when the stimuli were truly random (unpredictable) within the same context. (As a side note: for historical reasons we often refer to self-propelled repeating episodes as spatial, from the animals' perspective there is arguably very little 'spatial' in moving a structured horizontal wheel by his limbs; see Aronov et al. 2017 Nature or Pastalkova et al., 2008 Science).

First, we apologize for not describing the ball experiments in sufficient detail. In the revised version, we included the missing information. Importantly, air puff stimuli were administered pseudo-randomly by the experimenter without locking it to any sensory or temporal cue (for example by repeating with a fix inter-stimulus interval). As detailed in the previous point, changing predictability of the stimulus because of coupling its delivery to location did not influence the response of air puff-activated pyramidal cells (aAP-Pyr). Consequently, response of aAP-Pyr in the first lap in an air puff epoch was larger than on the ball as now demonstrated in Supplementary Fig. 2 (cumulative distribution of response magnitudes on panel d and comparison of durations on panel b). Because the first air puff within a stimulation epoch could not be unforeseen similar to the pseudo-randomly delivered stimuli on the ball, we think that the difference in the response on the disc versus on the ball cannot be attributed to difference in the predictability of the stimulus. Regarding the Reviewer's side note, we agree that for the animal, active movement in an environment is radically different from running head-fixed on a disc, especially because in the latter scenario synchrony of sensory information flow from external sources and from internal sensors reporting the animal's movement is disrupted (e.g. changing tactile and visual cues but stable vestibular input). On the ball, all but proprioceptive information is locked (the surface of the ball is untextured), thus the main difference may be the amount of inconstant sensory information available for the animal. However, this information flow is controlled by the animal's motor activity akin in case of free movement.

2. The authors present no convincing data to support their claim that the 'AP-cells' and the place cells represent separate, distinct populations. The proportion of place cells and air-puff responsive cells in pyramidal cells are 53% and 40%. Assuming independent distributions one may expect 21% of pyramidal cells to respond to air puffs and be a place cell, pretty close to the observed 26 %. This suggests that amongst pyramidal cells having a place field and responding to air puffs combine independently. The firing rate differences between air puff responsive and non-responsive cells are not evidence in this respect (distributions that may indicate clear separability of populations are not reported). Also, the anatomical segregation of the two populations is not demonstrated unequivocally (see my comments below).

Therefore, to refer to these cells as a sub-circuit seems far-fetched and a bit misleading (even in spite of the results of assembly analysis).

Thank you for raising this important issue and thus enabling us the clarification of our findings. First, connected also to the next point, we revised the nomenclature to disambiguate the different air-puff responsive pyramidal cell subgroups. In the original version, we used the term “air puff-responsive” equivocally for referring to air-puff activated neurons. In the revised version, we used a term composed of two parts: the first unequivocally specifies the type of response the given group of neurons exhibited (aAP: air puff-activated, iAP: air puff-inhibited, nAP: air puff-unaffected) whereas the second part corresponds to the type of neuron (Pyr: pyramidal, can be PC: place cell or nPC: non-Place cell; IN: interneuron). It is important that in the aAP subgroup, air puff-activation had to be i) transient and ii) followed the air puff after it was translocated (in some cases activation could be observed but did not reach significance, therefore such neurons were not categorized as aAP-Pyr). It is important to highlight that in case of inhibited pyramidal cells (iAP-Pyr, or iAP-PC and iAP-nPC) the response did not follow translocated air puffs. In the analysis of air puff effect on place-modulated firing (Fig. 6), only those place cells were used that received an air puff both outside their place field, and in a different air puff epoch, inside their place field. The majority of them did not satisfy the two criteria for being classified as aAP-Pyr or specifically, aAP-PC. In most of these neurons the statistically dominant response was suppression of activity, therefore most of these were classified as iAP-PC. To link air puff response categories and the neurons selected for this analysis, we have complemented Fig. 6a with this information.

We acknowledge that deciding if aAP pyramidal cells corresponds to a separate subgroup is unattainable without further in vitro electrophysiological, morphological or molecular evidence clearly beyond the scope of the current study. However, some basic characteristics (activity during movement and ripples and number of interneuron targets) distinguished them from inhibited or non-responding pyramidal cells. Probably, as we mentioned in the Discussion, these neurons’ inputs may explain both the effect of air puff on them and differences in their state-dependent activity. The possibility that aAP pyramidal cells form a subcircuit was motivated both by the higher number of interneurons these cells target and by the assembly analysis as pointed out by the Reviewer. However, because of the lack of further evidence, we removed the speculation about a subcircuit composed of aAP pyramidal cells.

3. I suggest the authors revise the terminology they use/introduce in this manuscript. By coining the term AP-cells the authors suggest that these cells ‘evolved’ to encode air puffs. While air puffs are very useful to apply mild aversive stimulation in a quantitative manner, their occurrence and relevance in natural habitats is not clear, and it is hard to imagine there are cells selective for this particular modality of aversive stimuli. It is clear these cells do not respond to painful tail shocks and neutral sensory stimuli. It would be nice to see if the response generalizes to stimuli of similar valence (mild aversive) but different modality.

Regarding the change of terminology, see the response to the previous point. We agree that it is improbable that the detection of air puff *per se* for what aAP pyramidal cells are evolved for. We rather think that converging inputs carrying multimodal sensory information characterizing an aversive stimulus (such as an air puff) distinguish these neurons from place cells. Different input patterns would be activated by other types of aversive stimuli depending on the sensory modalities dominating the actual stimulus. Thus, we think that aAP pyramidal cells are selective not for the air puff but for a constellation of sensory stimuli characterizing, among others, air puff stimuli. Activation of a mostly non-overlapping group of cells

by the tail shock supports this assumption. We inserted a remark about this possibility into the Discussion. We think that both a positive and a negative outcome of experiments aiming to test the effect of a different type of mild aversive stimulus on the response of air puff-activated pyramidal cells would support the above assumptions. Largely overlapping response to the contrasting stimulus would indicate large extent of convergence of a wide set of inputs on activated neurons whereas the absence of effect would point to more input specificity. Moreover, it would still not be possible to exclude that a third type of stimulus fails to activate these neurons. In light of our current results, the potential inconclusiveness and the time necessary for running these experiments and analyzing the data, we think our findings are sufficient for supporting our revised claims about the representation of aversive events in the dorsal CA1 without data from additional experiments.

In addition, the abbreviation AP is extensively used as for action potential in neurophysiology, which may confuse readers.

We thank the Reviewer for pointing to the potential confusion this abbreviation may cause. To make our nomenclature unambiguous, we replaced the “AP” abbreviation with the full expression (air puff) when used for the stimulus and in case of cell group names, we embedded it into two-component terms (see the point about revision of the nomenclature).

The authors use AP-responsive cells inconsistently. In general air puff response may be inhibition or excitation. The data presented shows that the air puff induces a rather complex response on the population level with some cells (in fact the majority) decreasing and others increasing their firing. There is no scientific justification why increases are more relevant than decreases.

Again, we would refer the Reviewer to our response about the nomenclature. Our intention was clearly not to rank the types of responses based on importance, and we apologize if we had made this impression in our Reviewers. We would instead emphasize the division of labor among the different components of the dorsal CA1 circuit as well as the intricate temporal organization of responses during the processing of the aversive event. We hope that in the revised version we could better communicate our findings and the conclusions we drew from them.

Finally, the term conjunctive AP-cells suggests that these cells respond to position and air puffs in a combinatorial way, like conjunctive HD-grid cells in the mEC. The authors do show interactions between responses, yet the response properties of these cells are not conjunctive in the strict sense.

We agree with the Reviewer and corrected the description of air puff-activated place cells. Instead of calling them conjunctive, we used the term “mixed tuning” that better corresponds to their feature selectivity.

It is more useful to report standard deviation than standard error as a data dispersion measure. Why did the authors chose otherwise?

We acknowledge that using standard deviation would be more informative. The reason of using s.e.m was rather habitual but in the revised version, we displayed wherever possible raw data points and box plots on figures and SD in the text. We showed s.e.m only for the mean PSTH for illustration purpose but we always indicated in the figure legend.

Specific comments:

Abstract, line 22-23: Rephrase and separate this sentence.

We have rephrased the sentence.

Abstract, line 25: The sentence implicates that place cells are responsive to non-salient stimuli, but the tone stimulation did not evoke cellular responses at all.

We removed the misleading claim.

p3, line 93; Please report statistics for the proportion of cells preserving their air-puff associated firing characteristics (negative or positive) over numbers of different air puff locations (e.g. 2/3, 3/3, 3/4 or 4/4).

We inserted a pie chart showing these proportions in Supplementary Fig. 1.

p3, line 106 There is a discrepancy between the text (reporting proportion of AP cells being not place cells) and the total numbers (given as all pyramidal cells). The reporting of data needs to be consistent throughout the manuscript.

We have corrected the numbers.

p3. The spatial distribution of different cell categories along the radial axis in the pyramidal layer is insufficiently described in the manuscript. The mean distance from the center of stratum pyramidale being around zero may mean uniform distribution along the radial axis as well as enrichment in the middle (as suggested by the authors). In addition, giving the SEM and not SD as dispersion measure, prevents the reader from inferring or even comparing the distributions. In this case also the ANOVA is inappropriate as the relevant question is whether the distributions are different and not if one group is on average more superficial or deeper than the other. I suggest to include the actual distributions of distances.

Thank you for the Reviewer's suggestion. We replaced the plot and on the new version raw data points overlaid onto a violin plot are shown. We also replaced ANOVA with its non-parametric version (Kruskall-Wallis ANOVA) for testing group-position interaction.

p4, line 143; Please provide statistical evaluation of this data considering that perhaps the number of interneurons firing on the trough and on the peak are also much different. Defining interneurons firing on the trough and on the peak is not trivial (there are other interneuron types firing on intermediate phases, and the within-group variability is also high). Please provide the details.

Thank you for pointing out this complication. We have complemented the analysis first, by calculating the average Z-scored response in each bin of the population phase preference histogram (we also presented the raw preferred phase versus response plot). Then we divided the interneuron population into four subgroups based on the phase ranges in Czurko et al., 2011, J. Neurosci., plotted the peri-air puff firing histogram of every interneuron in each category with their averages.

p4, line 143; Since the number of potential presynaptic and postsynaptic partners varies significantly across the different groups (multiplication further increases the differences in the numbers of potential contacts) comparing the numbers of detected functional interactions is not necessarily reflecting actual differences. Please use likelihood calculus.

Thank you for highlighting the potential confounds of this calculation. Importantly, we changed the detection method by replacing a hard threshold with a statistically robust variable threshold based on

(Stark and Abeles, 2009) that resulted in a significant increase of detected putative connections. Then, we used Kruskal-Wallis ANOVA for comparing the number of targeted interneurons across groups. We believe that this analysis convincingly reveals the differences among the air puff response groups.

p5, The last sentence of the § in lines 185 186 is incomprehensible.

We apologize for the wrong composition of this sentence. We clarified and simplified it.

p5, line 226-8; The authors claim that the phase precession in AP cells started at later and ended at earlier phases than in place cells. One may arrive to a conclusion that this expands the phase range covered during the phase precession, which is the opposite of what the authors actually report. In fact the interpretation of 'earlier' and 'later' is problematic with circular data. For example the start of the phase precession in AP cells is just after the trough in AP cells (29°) and just before the peak in place cells (147°), which is almost opposite phase of the cycle (which is earlier or later?). In contrast in the four representative examples of Fig 5, the difference in the starting phases in AP and place cells appears to be minimal. Please display the individual ranges in Fig 5, as these are not trivial from the start and end distributions (and is as informative as that). Furthermore the description of methods for determining start and end phases of phase precession is insufficient, and it is far from trivial how one derives these from the circular-linear regression (also the steepness of the position-phase relationship may not be stationary across the entire place field).

We have now replaced the start and end phase distribution graphs with one that shows the individual regression lines whereby the difference of start/end phase, slope and phase range of all aAP-Pyr versus nAP-PC is clearly visible. We also provided a more detailed description of the circular-linear regression used for the analysis of phase precession (the same method as described in Kempter et al., 2012).

p6; In the text (line 248-9) the authors claim that correlations within control place cell maps are higher than correlations between the maps recorded during stimulation and control including the pre vs. Post control sessions as well (text in brackets), but the legend of the referred Fig 6c indicates that it was pre-pre, ap-ap and post-post but not pre-post correlations compared to ap-pre and ap-post. Please clarify, and if pre-post correlations are not calculated, include them as well. A detailed description of place map calculus and how the above correlations were calculated is also missing from the methods section (if there was any smoothing applied, for example).

We have included the pre-post correlations in the revised version and provided a detailed description of both the place map calculation and about how correlations were determined.

p6; Ripple participation analysis should account for any variations in the baseline firing rate. The current calculus will not adequately measure specific ripple participation but will instead depend on other variables as well.

We have complemented the analysis of ripple participation by introducing an additional metric based on average number of spikes per ripple relative to spike number during inter-ripple immobile periods. This measure thus takes into account the mean level of neurons' activity in the brain state ripples are embedded in.

p6; The terms assemblies and patterns are used as synonyms. This is confusing, the two terms aren't interchangeable.

We changed “pattern” to “assembly” in the revised version of the manuscript except the artificial assembly template that we called “synthetic air puff-pattern”.

p21 methods, line 886 onwards; Please indicate if the z-scoring was performed over averages or binned firing rate changes and over what extent (length) and type (brain state, movement) of recording. Also, the reporting of firing rate changes is often not clear (what was averaged, peak or mean rates, etc.), and obscure abbreviations are used on axes (Norm. Sp. No. *10-3).

We have given a detailed description of the metrics and how they were calculated in the revised version of the Methods. Esoteric axis labels have also been replaced by clear-cut names of the plotted variables.

p22 methods, line 906; There are several well established methods in the literature to evaluate potential monosynaptic connections. The method used by the authors appears somewhat arbitrary, please provide the references and the rationale.

As briefly mentioned above, we changed the procedure of detecting putative monosynaptic connections. Instead of using the hard threshold, we utilized the method described in Stark and Abeles, 2009.

Fig 1c and S1a: The purple lines in the histograms are easily confused for a histogram data point. Please replace by a more distinct sign (an arrow for example); Peri AP firing histograms are not interpretable in control trials (where there is no air puff).

Purple lines have been changed to white arrowheads. We also renamed the firing histograms to tuning curves (firing histograms along the spatial axis).

Fig. S1e: I could not find the colour scale bar.

A color scale bar has been inserted on Supplementary Figure 7.

Fig 1h: I don't understand what is on the x axis. Is it the log₁₀ of the latency in s? – *shouldn't we change it?*

We have changed the legend to be unequivocal: it is log₁₀(latency of 1st spike in 1st significant post-air puff bin) as indicated by the Reviewer.

Fig 2b: The violet bars are difficult to discriminate from the colour-scaled image behind.

Purple lines have been changed to white arrowheads.

Fig. 6a: The label and units for x-axis is missing.

We put the labels.

Fig. 6c: The colours of the control and AP distributions are hard to distinguish. Please change.

The color has been changed for making it more discernible.

Reviewer #2 (Remarks to the Author):

Barth et al. adopt state-of-the-art behavioral and recording techniques to investigate if there is a subcircuit specialized for aversive stimuli in the hippocampus. The authors elucidate a cell population comprising both pyramidal cells and interneurons that selectively fire for an aversive air puff response. The firing of these cells is further modified depending on the availability of spatial information in the environment and the salience of the aversive stimuli. The authors then impressively show that these cells exhibit theta phase precession and participate in ripples, suggesting some kind of sequential coordination between these aversion-responsive cells that may guide behavior and memory consolidation. Lastly, the authors suggest that aversion is represented by pre-formed cell assemblies before even receiving an aversive stimulus. The authors also show through a decoding analysis that following a shock, the hippocampus increases its representation of the reward zones.

We thank the Reviewer for the meticulous assessment of our work and for his constructive comments and advices.

On a general level, the paper is interesting but my enthusiasm is muted by some unsupported claims, as well as somewhat careless and/or inconsistent presentation of abbreviations, statistics, and presentation. Statistics are also not presented altogether in some cases.

We would like to thank the Reviewer for the favorable opinion about our manuscript as well as for the helpful comments and suggestions. We apologize for the inconsistencies of data presentation, for incompleteness of statistics and for not describing the methods in sufficient detail. We have extensively revised the manuscript by clarifying nomenclature, complementing the Results with new analyses and statistics and we included several methodological details lacking in the original version.

Figure 4 somewhat implies that there is not a true aversion circuit in the hippocampus. And Figure 6 is interesting, but again suggests that aversion also influences non-AP place cells. This isn't too problematic, since it has been shown that spatial cells respond to all kinds of sensory stimuli. But it does bring into question how specific this "aversion circuit" really is for aversive stimuli.

We think that air puff-activated pyramidal cells are rendered selective for the applied aversive stimulus by the pattern of inputs conveying multimodal sensory information and activated by the air puff. It is possible that other, overlapping mild aversive stimuli would also activate these neurons. We agree that data presented on Fig4 and in the respective part of the Results, do not support the existence of a general aversive circuit in the dorsal CA1 but imply that minimally overlapping subgroups of neurons are tuned for divergent types of salient events. We speculate that event-specificity may be explained by the distinct patterns of inputs targeting these subgroups of neurons. We changed several parts of the manuscript to better reflect the aforementioned interpretation.

Figure 5 is impressive and interesting, showing that theta phase precession exists even for non spatial coding.

We agree, this finding may point to key aspects of how various types of codes are temporally organized in the hippocampus. Partly this finding motivated the assembly and decoding analysis.

The assembly analysis is interesting in concept, but (likely) suffers from such low sample size of available AP assemblies such that statistics cannot be run properly. The low number of AP assemblies (~7% of all detected assemblies) also does not reconcile the earlier results showing that a pretty significant

proportion of ALL hippocampal cells (>30%) can encode an air puff and/or a tail shock. It seems that the authors are making a huge stretch in their interpretation here and I would advice caution.

First, we have to apologize that we could not describe the motivation behind assembly analysis and its details in sufficient clarity. Partly based on the phase precession analysis we aimed to test if air puff-activated pyramidal cells (aAP-Pyr that we formerly termed AP-cells; 16% of pyramidal cells) tend to fire together forming an assembly even without the aversive stimulus. After finding out that this is the case, we uncovered that the assemblies collecting aAP-Pyrs are tightly coupled to the air puff but strikingly, they emerge before and independent from the aversive event. This suggested that there may be a “subcircuit” composed of aAP-Pyrs. However, because of the lack of further evidence supporting the existence of a subcircuit, we removed speculations about it from the manuscript.

We also acknowledge that air puff-elicited suppression may also be a key component of the dorsal CA1 circuit’s response to the aversive stimulus, but we focused on the activated (thus air puff-coupled) pyramidal cells. The main reason of it is that we think these neurons would transmit the aversive stimulus to the circuit and their activity may partly be responsible for the sequence of changes triggered by the event including the recruitment of inhibition, suppression of collocated place fields (the inhibition of pyramidal cells) and remapping. Because of the possible divergence of connections these activated pyramidal cells possess (as suggested by the analysis of monosynaptic connections), activation of one assembly composed mostly of these neurons can have profound effect on the network.

While the positional decoding analysis is interesting in terms of implications, those results do not tie together the main focus of the narrative presented through Figures 1-7 which were focused on aversion-responsive cells. The lack of a circuit-level explanation for why the decoding results are the way they are renders the last figure more questionable than insightful. I suggest that the authors try to search for a physiological explanation of the decoding analysis by further analyzing the results presented in Figure 6. Otherwise, the findings of this decoding analysis under shadow the significance of all the other figures because the decoding analysis counterintuitively suggests that this “aversion circuit” is not as important as the authors think.

We agree that this part of the analysis was a bit detached from the rest of the results. The logic behind this analysis was to uncover the consequence of all the single neuron-level changes we described in the preceding parts of the results on the code carried by the population. The result of this analysis was rather puzzling but remained valid after statistical validation. However, an explanation was still missing. For the revised version we carried out an additional analysis to uncover the cause behind this unexpected finding. In the original version we already demonstrated that air puff-activated pyramidal cells are not responsible for this phenomenon. Thus, we ran a new analysis in which we revealed that during the aversive events theta cycles are dominated by place cells with place fields close to the reward zone simultaneously with a slightly depressed activity of place cells with place fields further from the reward zone (see also Figure 6 and corresponding analysis regarding the suppression of air puff overlapping place fields), consequently position estimation located the animal close to the reward zone. This analysis may proved a clue why the place code shifts to the reward zone. It is still a question what the reason of the activation of reward-proximal place cells is. Because this analysis was run on the air puff epoch located most distally from the reward zone, heightened excitability of the reward proximal place cells can be excluded as an explanation. We think, testing further possibilities would be beyond the scope of this paper but can be a possible direction of future studies.

Figures 7 and 8 are the most problematic for me. In many ways, it feels like the authors are overly ambitious in trying to link their prior results throughout Figures 1-6 in a greater context. As such, both figures appear disconnected from the remaining narrative.

By more analysis and clarification, we hope these parts are more integrated with the rest of the manuscript.

Major Comments:

Figure 1:

- No supplementary material is shown, making it impossible to review claims that refer to them.

We apologize for this mistake but during submission we had uploaded both the main manuscript and the supplementary material, therefore we do not have an explanation about its omission.

- Figure 1f: There are 16% of cells that decrease activity and are place cells. By definition, place cells should be spatially selective and silent when the animal is not in the place field. Since the air puff was administered at a minimum of 3 different spatial locations to ensure non-overlap with the place field, it is curious that a place cell could suppress their (already silent) firing even more following an air puff. Why is this the case? And as an aside, it would help to show individual cell examples of this kind of cell, as well as other kinds of AP cells (those that may have overlapping place fields or cells that are not as clean in response) in the supplementary.

We thank the Reviewer for drawing our attention to this potential discrepancy. This subgroup of pyramidal cells mostly corresponds to the subgroup in which place cells had a place field overlapping with an air puff location (see also Fig. 6). In these cases, the dominant response was inhibition (in the collocated place field and no effect or non-significant activation outfield). These neurons were further analyzed when comparing the effect of the aversive stimulus on infield versus outfield activity (Fig. 6). In the revised version of Fig. 6a we included which response category these neurons belonged to. It should also be noted that some non-place pyramidal cells were also inhibited (see Supplementary Fig. 1c). In these cases, place fields could not be identified based on our detection criteria (now given in greater details in Methods) but activity on the disc was high enough to enable the observation of inhibitory response to air puff.

- Figure 1j: This is hard to reconcile given that in Figure 1f, there are more AP cells that reduce their firing following an air puff (24%) as opposed to those that increase their firing following an air puff (16%). On another note, it is not clear what cell population is being shown in this panel. Is it all AP cells? Also, the figure caption indicates that a 2-way ANOVA was used. But the panel only groups the data by location (pre, AP and post). The second factor of “stimulus” is not shown. This is confusing because it seems a 1-way ANOVA could’ve sufficed for this panel.

We apologize for the confusion that may stem in the inconsistency of our original nomenclature. We introduced a nomenclature that hopefully makes the presentation of the data clear. We no longer use the term “AP-cell” but refer to the specific response and cell type (e.g. air-puff activated pyramidal cells are now aAP-Pyr). We provided a list of definitions of terms in the Methods. On the original fig1j aAP-Pyrs were shown but we removed this panel and inserted a new graph in SFig1f that shows the firing rate of this subgroup of cells at the 3 air puff locations. On this graph error bars correspond to SD instead of SE.

- The two panels in Figure 1k should be combined into a 2-way repeated measures ANOVA, the repeated factor being movement (immobile or moving) and non-repeated factor being cell type (AP or non-AP). Then, follow with a significant interaction effect, appropriate repeated measures correction and post hoc tests. If a repeated measures design is not appropriate for whatever reason (such as missing data), then the stat design should be at the very least be a regular 2-way ANOVA followed by a significant interaction effect and appropriate post hoc tests.

Thank you for the Reviewer's suggestion. We have revised the statistics and used the 2-way repeated measures ANOVA and updated the figure and its legends. We also show the four major categories of responding pyramidal cells (activated and non-affected place and non-place cells, Fig. 1j). We present inhibited pyramidal cells together with the abovementioned subgroups on an additional graph on Supplementary Fig. 3b with the accompanying statistics. We also changed the mode of presentation to box plots with overlaid raw data points.

- As it currently stands, given the nonsignificant result, it is inappropriate to state in the text that a certain population of pyramidal cells is more superficial than another.

We agree with the Reviewer's remark, therefore we revised the graph to better demonstrate the position of all the recorded activated and non-affected pyramidal cells. Now we show the individual data points combined with a violin plot in the new version of this panel. Despite the lack of significant difference among pyramidal cell response groups, we still think it important to mention the bias we found in the distribution of positions in the pyramidal layer.

Figure 2:

- Figure 2b: Second interneuron example is not convincing. Is there a better example?

We have changed the example with a more convincing one.

- Figure 2f: Are biphasic interneurons consistently biphasic? I worry that the biphasic activity is just spike contamination from the interneuron's natural high firing rates that mixes with suppressed activity following an air puff.

We have identified the two phases based on thresholds detailed in the Methods. According to these, the initial activation was defined as spike count exceeding pre-air puff mean by 2SD (never embedded in the inhibited period). We think that contamination would not have resulted in a peak that overshoots that threshold within a short temporal window after air puff onset.

- Figures 2i-k: Doesn't a monosynaptic connection imply that the interneuron could be synapsing onto the pyramidal cell as well?

We agree with the Reviewer that it is rather the case. However, because of the low firing rate of the pyramidal cells relative to interneurons, inhibition is much more difficult to identify and statistically verify. Therefore, we focused on the excitatory connections.

- Despite no significance in Figure 2j, the authors make an unsupported claim: "AP-cells were more prevalent among the presynaptic partners of AP than of non-reacting interneurons".

We have toned down this claim by indicating that this finding was just a tendency not reaching significance.

Figure 3:

- Statistics are not reported for Figures 3c and 3g.

We have provided the statistics in the revised version.

- Lines 173-174: “Thus, AP-triggered interneurons tended to be modified in the opposite direction by the cue rich to nonspatial context switch.” This statement is inaccurate since facilitation and suppression are BOTH enhanced by the air puff in pyramidal cells. There is no rationale here to suggest a story in which the interneuron population is affected in the “opposite” direction since there is no opposite direction to begin with.

We have removed this claim.

Figure 4:

- Figure 4 brings into question whether there is really a hippocampal subcircuit for aversion. This figure shows that air puff cells for the most part are unrelated to tail-shock cells. In other words, AP cells don't serve a general role for aversion (or at least, only a very small percentage do – 8%).

We acknowledge that raising the possibility of an aversion-specific subcircuit was overinterpretation of our data despite some of our findings may point in such direction. In the revised version we would rather emphasize that tuning to a given type of aversive event may be determined by the inputs the triggered neuron receives and different types of events activate only partially or non-overlapping subpopulation of cells. These neurons then would be key for resculpting hippocampal representations.

- The counterargument is that these results is sort of akin to global remapping of place cells between contexts. Some place cells are silent in one environment, whereas they are active in another. Perhaps this is what is taking place here in the context of aversion. However, it seems a little unlikely to me, given that the spatial context itself is unchanging. (And, we saw from Figure 3 how much influence the spatial context exerts on AP cells.) And furthermore, the fact that you then have another subset of cells that is specifically responsive only to the reward makes it more likely that these “AP cells” are not really part of an aversion subcircuit but serve a broad hippocampal function to store the animal's experience, or salient stimuli in general.

We can agree with the Reviewer regarding the potential function of these neurons. It is fully possible that, as we also pointed out in the Discussion, they serve to code the aversive event and then facilitate its integration into memory by forming sequences into which place and non-spatial codes are amalgamated.

Figure 5:

- Figure 5: Do conjunctive AP cells have the same phase precession range for both the AP stimulus and a normal place field? Or is the phase range greater for the place field only?

We thank the Reviewer for this important point. Below is the comparison of the phase range covered by spikes of phase precessing mixed-tuning pyramidal cells (aAP-PC) during air puff delivery and in their place field. There is a tendency of larger phase range of place field spikes but probably because of the low neuron number and large variability the difference is not significant.

- Figure 5: There is no way to determine if recruitment of conjunctive AP cells during ripples is for consolidation of the AP (aversion) or place fields (space). It is recommended to run this analysis for pure AP cells and see if they participate more during ripples than non-AP cells.

In the revised version we extended the ripple analysis by quantifying it using an alternative metric: spike count in ripples relative to spike count in ripple-devoid immobile periods. We also analyzed the ripple-coupling of all response and pyramidal cell type categories. These new results are shown on Fig. 5e,f and Suppl. Fig3c.

Figure 6:

- Line 242: “Overall, in-field firing of place cells was suppressed if the AP was co-located with the place field whereas their activity”. This statement is not entirely correct given that about 10-15% of place cells seem to increase their firing following an air puff. I would suggest modifying the sentence to make it more conservative.

We have modified this sentence.

- Continuing off the same point, could it be possible that some cells had an air puff towards the end of the cell’s place field? In such a case, the suppression of activity is natural, and not due to the AP.

We would like to thank for raising this very important concern. We have made an additional analysis, results of which are shown on Suppl Fig. 8a,b and detailed in the Results. We determined the distribution of relative location of air puff onset and its relationship with the response. The distribution was close to uniform whereas no correlation could be detected between infield air puff onset location and response to the stimulation.

- Line 243 and Figure 6b: First is not clear why a z-score less than 2 is nonsignificant. No statistical explanation is given. Second, if one were to run a statistical test, it seems to me that the best approach here is to compare pre-AP response versus post-AP response using a repeated measures test like the sign rank test.

Any values exceeding 2SD would lie outside 95% of the distribution. This threshold has been used for categorizing an effect as significant (for its use, see for example Carcea et al., 2017 or Klee et al., 2021, also listed as Refs 49 and 50 in the References).

- No stats are provided for the cumulative distribution graph in Figure 6c. Also, please use different colors/line markers to differentiate different comparisons. There is no way to differentiate which line is which comparison.

We agree, the original form of this graph was not informative, thus we replaced it by the box plot with statistics corresponding to correlation values on panel e.

- Difficult to assess the merits of Figure 6 without access to the supplementary figures.

We hope the Supplementary information will now be available with the revised manuscript.

Figure 7:

- Can the authors provide a layman explanation of the assembly analysis for readers that are unaware of the paper that the analysis is based on?

We have prepared an explanatory figure. Unfortunately, it was seemingly missing from the version of submission our Reviewer received. In the revised version, we complemented this figure with additional panels illustrating the assembly detection process.

- Figure 7a: This figure looks more like an example of a single cell rather than an assembly.

The panel shows the position-trial raster of an air puff-coupled assembly. The tight coupling of this assembly to the air puff is striking.

- Figure 7b: How can an assembly really be an assembly if the population is 2 or less?

We apologize for not being clear enough in the legend. This panel shows the number and weight of aAP-Pyr cells in air puff-coupled (AP-a) and air puff-independent (nAP-a) assemblies. Each dot corresponds to an assembly and not to individual neurons. Larger number and weight of aAP-Pyr cells in air puff-assemblies is nicely visible.

- Figure 7e: A two-way ANOVA should be used here. Also, in the figure caption, please provide description statistics.

We used two-way repeated measures ANOVA in the revised version. Both the median, interquartile and 10-90% range and individual data points are shown on the plot.

- Lines 296-299: "We were able to identify pre- and post-session ensembles resembling the AP-patterns but the weight of AP-cells in these patterns was significantly lower in most of the cases than in similar AP-assemblies isolated during the AP session". This is a false statement given that the panel shows non-significance.

We apologize for the lack of clarity. While we could identify significant effect of condition (pre, disc, post) by 2-way repeated measure ANOVA, post hoc tests failed to reveal significant differences between pairs of conditions possibly due to low number of air puff assemblies and high variability of their strength. We inserted the description of this finding into the assembly analysis part of the Results.

- Figure 7g: No post-hoc tests are carried out between the purple and black bars, or between the purple bars. Since the argument in this figure is that assembly strength increases, statistical significance should be shown to support this claim.

We have revised the statistics: a two-way repeated measures ANOVA followed by Tukey post-hoc test was used and its results were included in the legend.

- My general worry in this figure is aside from the fact that assemblies have such little number of cells, the actual number of AP assemblies is also extremely low (only 7 versus 84 non-AP assemblies). The low assembly population and count do not really reconcile the results presented in Figures 1, 2 and 4 where >30% of cells have some kind of AP response. It therefore seems like a stretch to say that representation of an aversive event is allocated to a pre-formed aversion assembly.

The number of significantly contributing neurons (with large enough weight defined in the Methods) in an assembly can vary between 2 and 8 (in our case, see for example Supplementary Figure 9b; it should be noted that it can be higher with more simultaneously recorded neurons). Detecting only one (maximum 2 in one session) air puff-coupled assembly in a session indicates that air puff-activated pyramidal cells (thus, 16% of all pyramidal cells) are prone to form assemblies by preferring to fire together. Temporal synchronization of their activity within assemblies can amplify their effect on downstream targets. Additionally, the connection between these neurons and the inhibitory circuit is more divergent than that of non-affected pyramidal cells raising the possibility that a small number of these cells can have profound effect on the hippocampal network (it should be noted that the analysis of monosynaptic connections underestimates the number of real connections). Their output can be broadcasted to an even larger population outside the hippocampus, the study of which is well beyond the scope of this study.

Figure 8:

- Figure 8c: Please make sure the y-axis scale bar ranges are the same and report descriptive statistics (such as mean and s.e.m. or std) for the decoded results.

We have corrected the y-scale according to the Reviewer's suggestion.

- Figure 8 is potentially very interesting, but it falls short without an explanation for why the air puff reconfigures the place map to encode the reward areas more. For instance, Figure 8 suggests that this shift in place map has nothing to do with AP cells. Therefore, the question is: "what is the circuit-level mechanism that explains why the air puff is causing the change in decoded location". If the conjunctive AP pyramidal cells aren't responsible, then the answer must lie in the non-AP cell population. Logically, if there is an increase in decoding of reward location instantly following an air puff, then it suggests perhaps one of two explanations:

1. The training data of each air puff epoch is "the second half of the preceding control epoch" (Line 322). Since these control epochs are interweaved between air puff epochs, it is possible that there was place

cell remapping (as a result of air puff; Figure 6) during these control epochs. Is this true? Can it be shown that place cell remapping in Figure 6 was more biased towards the reward zones? For example, in Figure 6a, in the “outfield” graph, after the air puff it seems like there is an increase in spiking in many cells. Therefore, the increase in reward representation in place cells is due to increased remapping of non-AP place cells towards the reward zones.

We agree with the Reviewer, this is a very interesting and rather puzzling finding. For training the decoder we used the 2nd half of the first control epoch for estimating the position during the first air puff epoch and the 2nd half of the first inter-air puff control for the 2nd air puff epoch. However, the analysis generated the same outcome: shift of representation to the reward zone during both the first (when the animal was naïve) and the second air puff epoch. Additionally, despite extensive remapping, the distribution of place fields along the animals’ path after air puff delivery was almost uniform thus not any part became overrepresented (Supplementary Fig. 8e). Thus, the representational shift to the reward zone was transient and could be detected only during the air puffs and lasting drift of representation to the reward zone that may have been biased the position estimation could neither be observed.

2. An alternative reason for increased reward coding is that following an air puff, there is an increase in activity of place cells whose natural place fields are the reward zones. Again, in Figure 6a, the increased activity of some cells in the “outfield” graph might belong to those place cells with reward zone place fields.

By additional analysis, we could verify this insight of the Reviewer: theta cycle population vectors during air puff delivery that coded peri-reward locations were dominated by place cells which had a peri-reward place field. Or, as the Reviewer raised: “...following an air puff, there is an increase in activity of place cells whose natural place fields are the reward zones.” We included this new result in the revised version of the manuscript. It should be noted, the effect of these reward-proximal place cells may have been amplified by the suppression of air puff overlapping place fields shown on Figure 6.

- But also, regardless of which of the above scenarios is true, Figure 8 makes it unclear just how important this “AP cell/aversion subcircuit” is for reconfiguring the place map towards positive valence stimuli. If anything, the results argue that there isn’t a true subcircuit for aversion, but the place code in general has the capacity to reconfigure its responses in the face of aversive stimuli.

As we described earlier, instead of a general aversive circuit, there are neurons tuned for a constellation of sensory inputs signaling an aversive event. One such population may be sensitive to input patterns activated by an air puff (or similar stimuli). These neurons recruit inhibition (possibly feedback inhibitory neurons in largest proportion) resulting in the suppression of place fields overlapping with the aversive stimulus. This mechanism would trigger (or facilitate) the reconfiguration. As described in the Results, the shift of decoded position to the reward zone was only transient and after the last air puff, we could not detect the overrepresentation of the reward zone and increase of place field density in the aversive segment could not be detected either, despite the massive remapping of many place cells.

- Lastly, what if the reconfiguration is not biased towards reward or positive valence? The reward zones were the only other salient locations in the task. What if the reconfiguration is really biased towards any other non-air puff location in the environment that has some basic level of salience?

It is a plausible alternative. However, after the first air puff epoch there are two salient locations that may have decreased the probability of a representational drift to the reward zone. Additionally, the second air puff location was closer to the reward area than the first, therefore the excitability of place cells with peri-reward place fields may have been different than during the first air puff epoch. Despite these two factors potentially in play, the probability of positioning the animal to the reward was decreased only moderately (insets on Figure 8c, left plots).

- Figure 8 decodes air puff epoch locations using the preceding interweaved control epoch. It is unclear what “changes” have occurred to neural activity during these control epochs as a result of preceding air puff epochs. Do the results presented in Figure 8 hold true if air puff trials are decoded using ONLY the pre-AP control epochs as training data? The pre-AP control without any air puff serves as the true baseline control.

We analyzed remapping between the different controls and air puff epochs (i.e. 1st control vs. 1st air puff, 2nd control vs. 2nd air puff, etc.) and despite extensive remapping (both shifting of place field locations and emergence / vanishing of place fields), we could not detect the overrepresentation of any segment of the disc (see Supplementary Fig. 8d and f).

Minor Comments:

- Figure references are not in order. For example, Figure 1c is referenced before Figure 1d on line 86. Or, when Figure 7f is referenced before Figure 7e. On a related note, sometimes figures are entirely unreferenced such as Fig. 1j.

We apologize for the inconsistencies in references for figures. We corrected these mistakes and hopefully now the order of figure references is correct and there are no orphan figure panels.

- Figure 1: In regard to selection of AP cells from the methods: “AP cells were defined as putative pyramidal cells with responses exceeding 2 z-score values for at least 50 ms duration in a 3 s long time window following AP and these responses could be detected at 2 or 3 out of 3 or 4 AP-locations.” Why do AP cells not all respond at all 3 or 4 of the air puff locations?

In few cases air puff-triggered increase of activity could be detected but it did not reach the 2SD significance threshold.

- Figure 1: Please define “sp” in the figure caption.

We removed this abbreviation and spelled out “spike: in every occasion.

- Figure 1 caption at line 467 uses the AP abbreviation before defining it.

We have changed it to air puff.

- Figure 1 caption at line 476: “Percentages” should be written in place of “Ratios”.

We have changed accordingly.

- Figures 3b and 3f show proportions, but percentages are reported in the figure caption.

We have revised the figure.

- Figure 4c shows percentages, but the caption refers to the data as “ratios” and the reported data are provided as raw counts. Please stick to one type of reporting data.

We have changed to percentages on the panel and provided the cell numbers in the legend.

- Figure 4: Please define TS and RW in the figure caption before using these abbreviations.

We have corrected the legend with the definitions.

- Figure 4: Were the shock, tone and reward administered in the same locations as the air puff?

We apologize for not clarifying this detail: different stimuli never overlapped. This information is now provided in the “Multichannel electrophysiological recordings” section of the Methods.

- Figures 4b, 4e: For all axis labels, only the first word should be capitalized. Figure 4b x axis is “Peri event time”. But Figure 4e x axis is “Peri Stim Time”.

We have changed the label.

- Significance stars between figures are inconsistent. For example, Figure 3b uses a single star for very small p values. But then Figure 5 uses two stars for similar p values. Please adhere to a single reporting standard such as: *: $p < 0.05$, **: $p < 0.01$, ***: $p < 0.001$. And please indicate this reporting criteria in every figure caption. (I.e., at the end of each caption, write “For all panels, *: $p < 0.05$, **: $p < 0.01$, ***: $p < 0.001$ ”). We used only $*p < 0.05$ and $**p < 0.01$ and also provided p-values wherever possible. • Figure 5: In the panels, please indicate that phase unit is degree, and ripple participation is percentage.

We have indicated it in the axis labels. In case of ripple participation, we used proportion.

- Figure 6c: Please indicate units for distance.

We have done so.

- Figure 8a: Spelling error: “Real positon”.

It has been corrected.

REVIEWERS' COMMENTS

Reviewer #1 (Remarks to the Author):

The authors have adequately addressed all my concerns and questions, and have significantly improved the manuscript, which I recommend to accept for publication. I have only a few smaller remarks left:

1. Results, line 149-151; This sentence should be toned down to sound more objective and neutral; the theta phase preference of interneurons does not define functional interneuron groups. Also the sentence below (line 160; „Thus, the aversive stimulus mostly activated...“) should be modified. Besides the fact that (in my opinion) it would be difficult to classify any interneuron as feedback and feedforward, there is no evidence that the aversive stimulus had activated any dendrite targeting or feedback interneurons (all activated cells may have been basket cells as well).

2. Results, line 169; please delete either ‚were‘ or ‚tended‘

3. Results, line 241-244. Besides my comments in the original review on problems with defining temporal relations (earlier vs. Later) in circular datasets, this description is still confusing and does not match one's observations in Figure 5. In the Figure 5b I see that in aAP-Pyr cells the phase precession starts much earlier (to use the author's terminology) than in place cells and ends a bit earlier, which indeed results in a reduced range covered. I suggest the authors use a description that refers to where phase precession starts and ends exactly in both response classes (e.g. precession from early ascending to late descending phase in aAP-Pyr cells).

4. Fig. 7 legend: the authors use ‚reactivation‘ for both assembly activation and true reactivation (such as during ripples).

5. Fig. S1 legend, line 6(2); I guess ‚middle‘ should read ‚right‘.

6. Please make sure that the text in all figures is of readable size. In particular, please enlarge figures S1, S2, S3, S7, S8 and S9. For Fig. S1d I had to employ a magnifying glass, which is not necessarily part of a reviewer's toolkit.

We would like to thank the comments and suggestions of our Reviewer. Our point-by-point responses are detailed below.

1. Results, line 149-151; This sentence should be toned down to sound more objective and neutral; the theta phase preference of interneurons does not define functional interneuron groups. Also the sentence below (line 160; „Thus, the aversive stimulus mostly activated...“) should be modified. Besides the fact that (in my opinion) it would be difficult to classify any interneuron as feedback and feedforward, there is no evidence that the aversive stimulus had activated any dendrite targeting or feedback interneurons (all activated cells may have been basket cells as well).

We acknowledge that the data at our disposal is insufficient for classifying the putative interneurons as feedback or feedforward. Therefore, we changed the phrasing and stated only the phase preference of interneurons in the Results. We reserved our speculation about the potential identity and function of the recorded interneurons for the Discussion.

2. Results, line 169; please delete either ‚were‘ or ‚tended‘

We have deleted ‚were‘.

3. Results, line 241-244. Besides my comments in the original review on problems with defining temporal relations (earlier vs. Later) in circular datasets, this description is still confusing and does not match one's observations in Figure 5. In the Figure 5b I see that in aAP-Pyr cells the phase precession starts much earlier (to use the author's terminology) than in place cells and ends a bit earlier, which indeed results in a reduced range covered. I suggest the authors use a description that refers to where phase precession starts and ends exactly in both response classes (e.g. precession from early ascending to late descending phase in aAP-Pyr cells).

Thank you for this comment, we have followed the suggested phrasing.

4. Fig. 7 legend: the authors use ‚reactivation‘ for both assembly activation and true reactivation (such as during ripples).

We changed reactivation to activation in the legend of Figure 7.

5. Fig. S1 legend, line 6(2); I guess ‚middle‘ should read ‚right‘.

We changed ‚middle‘ to ‚right‘.

6. Please make sure that the text in all figures is of readable size. In particular, please enlarge figures S1, S2, S3, S7, S8 and S9. For Fig. S1d I had to employ a magnifying glass, which is not necessarily part of a reviewer's toolkit.

We apologize for the difficulty of reading some of the texts. We have taken extra care of enlarging too small texts.